# Learning Dynamical Characteristics with Neural Operators for Data Assimilation

### Abstract

Data assimilation refers to a group of algorithms that combine numerical models of a system with observations to obtain an optimal estimation of the system's states. In domains like earth science, numerical models are usually formulated by differential equations, also known as prior dynamics. It is a great challenge for neural networks to properly exploit the dynamical characteristics for data assimilation, because first, it is difficult to represent complicated dynamical characteristics in neural networks, and second, the dynamics are likely to be biased. The state-of-the-art neural networks emulate traditional approaches to introduce dynamical characteristics by optimizing an objective function in which the dynamics are inherently quantified, but the iterative optimization process leads to high computational cost. In this paper, we develop a novel deep learning framework with neural operators for data assimilation. The key novelty of our proposed approach is that we design a so-called flow operator to explicitly learn dynamical characteristics for reconstructing sequences of physical states. Numerical experiments on the Lorenz-63 and Lorenz-96 systems, which are the standard benchmarks for data assimilation performance evaluation, show that the proposed method is at least three times faster than state-of-the-art neural networks, and reduces the dynamic loss by two orders of magnitude. It is also demonstrated that our method is well-adapted to biases in the prior dynamics.

## 1 Introduction

Data assimilation can be essentially defined as a statistical technique to combine the prior dynamics with a sequence of noisy and irregularly-sampled observations, which plays an important role in utilizing observational data, especially for numerical weather forecasting systems. For instance, the European Centre for Medium-Range Weather Forecasts (ECMWF) employs the variational assimilation algorithms in its operational systems to take full advantage of both in situ and satellite-derived data, as well as state-of-the-art numerical models (Rabier et al., 2000).

**Classical Assimilation Methods** Classical data assimilation methods range from early empirical analysis (Bergthorsson et al., 1955) and optimal interpolation (Gandin, 1963) to later variational-based assimilation algorithms (Sasaki, 1970) and the filtering methods based on statistical estimation theory, such as Kalman filter (Welch et al., 1995), and particle filter (Carpenter et al., 1999). Although classical methods have been essential tools for improving the capability of major numerical weather prediction centers worldwide (Kalnay, 2003), they generally do not account for the propagation of observation information along time and therefore are unable to effectively utilize observations at different times (Song et al., 2017).

**Time-dependent Assimilation Method** In recent years, new data assimilation methods have been developed (Evensen, 2003; Lorenc & Rawlins, 2005; Hunt et al., 2007). Unlike previous methods, they consider the revolution of observation information along time, in other words, the time-dependent dynamical characteristics (Song et al., 2017). Among them, the four-dimensional variational (4D-Var) assimilation algorithm is a cutting-edge one. Experimental results prove the advantage of 4D-Var assimilation methods in utilizing observational information (Lorenc & Rawlins, 2005). However, due to the complicated modeling and solving process of the 4D-Var algorithm, it is considered computationally expensive, especially for high resolution cases (Fisher, 1998).

**Machine Learning for data assimilation**     Prior to 2020, extensive research into developing machine learning for data assimilation had been successful in reducing computational costs (Hsieh & Tang, 1998; Vijaykumar et al., 2002; de Campos Velho et al., 2002; Härter & de Campos Velho, 2008; Cintra & de Campos Velho, 2018; Mack et al., 2020). However, these works base their theory on classical assimilation methods, thus suffering from low accuracy brought by not considering time-dependent dynamical characteristics. From 2020, a lot of neural network-based methods consider time-dependent assimilation. They can be generally divided into three categories: learning the inverse observation operator, learning model biases, and learning the optimization algorithm.

**Learning Inverse Observation Operators**     The aim of learning inverse observation operators is to use neural networks to construct a mapping from observations to reconstructed states. The representative work is Frerix et al. (2021), in which the observational data is exploited by a neural operator to provide better initial states for the optimization of 4D-Var objective function. This work has at least two flaws. The first is that the neural network is used only to exploit observational information, while the integration of observation and dynamical characteristics is still implemented by 4D-Var, leading to a limited improvement in computational efficiency. The second is that the strict constraints of time-dependent dynamical characteristics are the prerequisites for this work, which prevent it from generalizing to the cases in which the prior dynamics are biased.

**Learning Model Biases**     As mentioned above, the prior dynamics can be biased, and another group of works uses neural networks to learn the bias of the prior dynamic models for data assimilation (Brajard et al., 2020; Arcucci et al., 2021; Farchi et al., 2021; Bonavita & Laloyaux, 2020). These works generally achieve a balance between accuracy and efficiency, but the improvement in computational efficiency is insignificant, because the assimilation of dynamical characteristics and observations is still done in an iterative manner.

**Learning Optimization Algorithm**     Since the main computational overhead of the 4D-Var algorithm lies in the process of iteratively optimizing the 4D-Var objective function, some studies have focused on replacing traditional optimization algorithms with neural networks. Among them, the representative one is Fablet et al. (2021b;a). This work simultaneously learns both the dynamical characteristics of the model and the optimization algorithm of the 4D-Var objective function through neural networks. It not only takes into consideration the weak constraints of time-dependent dynamical characteristics, but also successfully reduces the number of iterations compared with traditional 4D-Var algorithms, subsequently improving computational efficiency. We consider this work to be the state-of-the-art method for comparison. Despite the achievements of the work, there is still room for improvement in terms of both accuracy and efficiency.

**Our Contributions**     We develop a new approach of combing data-driven strategies with model-driven methods for time-dependent data assimilation. This framework consists of three operators implemented by neural networks, namely, the inverse observation operator, the perturbator, and the flow operator. Among them, the cooperation of the perturbator and the flow operator decouples the two goals of learning observations and learning dynamical characteristics, and they succeed in continuously refining reconstructed states.

- To our knowledge, this is the first work that directly uses neural networks to blend the dynamical prior with a sequence of observations without explicit formulations of the 4D-Var objective function for assimilation.
- We test the proposed framework on the Lorenz-63 and Lorenz-96 systems. Experimental results support the effectiveness of our framework. The proposed architecture is at least three times faster than the state-of-the-art neural network Fablet et al. (2021b). It reduces the dynamic loss by two orders of magnitude and is comparable to the state-of-the-art neural networks in terms of the reconstruction error.
- We design experiments in which the prior dynamics are different from ground truth to simulate real-world scenarios. Experimental results prove that our method is well-adapted to deviations of the prior dynamic model.

## 2   PROBLEM STATEMENT

Suppose that there is a dynamic system with $d$ dimensions to be observed and estimated, and we denote the state variables by $\mathbf{x}(t) \in \mathbb{R}^d$. Consider $N$ evenly distributed time points, $t_i = t_0 +$

$i\Delta t, 0 \leq i \leq N - 1$, where $\Delta t$ is short enough for the integration step. We are interested in the assimilation problem over a fixed-length period, $[t_0, t_{N-1}]$, and we call the period of interest "assimilation window" (Tr'emolet, 2006; Fablet et al., 2021b). Our goal is to estimate the states of the system on these $N$ time points, $\mathbf{x}_i = \mathbf{x}(t_i)$. All the information we have is (1) the a priori knowledge of the dynamics and (2) the observations of the system states during this period. We note that in our work, the guess of the initial states $\mathbf{x}_0$ is not considered, and we focus on the integration of the prior knowledge and the observational information, as done in Frerix et al. (2021) and Fablet et al. (2021b).

**The Prior Knowledge**    In real-world scenarios, scientists develop physical models to estimate the dynamic system. We regard the physical model as the prior knowledge, and name it "the prior dynamics". The prior dynamics are usually represented by the following differential equation,

$$\frac{\mathrm{d}\mathbf{x}(t)}{\mathrm{d}t} = \mathcal{M}_{pr}\left(\mathbf{x}(t)\right), \tag{1}$$

where $\mathcal{M}_{pr}$ is a mapping from $\mathbb{R}^d$ to $\mathbb{R}^d$. The prior dynamics can be either biased or unbiased.

**The Observations**    The observations are indispensable for the reconstruction of the physical states. In our problem setting, the observations are assumed to be discrete and partial, with white Gaussian noise. This assumption is widely considered in the data assimilation community, especially for developing novel data assimilation algorithms (Brajard et al., 2020; Farchi et al., 2021; Arcucci et al., 2021). Suppose that all the observations are obtained at time $t_j = t_0 + jR\Delta t$, $j = 0, 1, ..., N/R - 1$ and only variables whose indexes belong to $\Omega_{t_j}$ are observed at time $t_j$. The formulation can be written as

$$\mathbf{y}_p(t) = \mathbf{x}_p(t) + \epsilon(t), \forall t \in \{t_0, t_0 + R\Delta t, ..., t_0 + (N/R - 1)\Delta t\}, \forall p \in \Omega_t \tag{2}$$

where $\epsilon(t)$ represents the white Gaussian noise.

## 3 LEARNING FRAMEWORK WITH THREE OPERATORS

The major difficulty of the problem lies in how to properly utilize the prior dynamic model. Numerous studies have developed methods based on Bayesian inference to deal with the time series states estimation problem by formulating the transition process in a probabilistic manner (Krishnan et al., 2017; Karl et al., 2016; Marino et al., 2018; Kingma & Welling, 2013). Despite their success, it is still difficult to properly deal with the biases of the prior dynamics for data assimilation.

Our work approaches this issue from a different angle. Instead of explicitly formulating the transition process, we simultaneously generate all physical states within the time window at once by an inverse observation operator. The prior dynamics are fed into our model via a flow operator that follows behind, which works with perturbers to progressively refine the reconstructed state. The overall structure of our framework is shown in Figure 1.

### 3.1 TWO METRICES

Before proceeding to the detail of our framework, let us first illustrate two important metrics for training and evaluation.

**Reconstruction Loss $\mathcal{L}_{rec}$**    The reconstruction loss is defined as the mean square error between the generated sequence of states and the ground truth states. Denote $\mathbf{x}_{gt} = [(\mathbf{x}_{gt})_0, (\mathbf{x}_{gt})_1, ..., (\mathbf{x}_{gt})_{N-1}]$ the discrete sequence of true states, $\hat{\mathbf{x}} = [\hat{\mathbf{x}}_0, \hat{\mathbf{x}}_1, ..., \hat{\mathbf{x}}_{N-1}]$ the sequence of generated states, then

$$\mathcal{L}_{rec}(\hat{\mathbf{x}}) = \|\hat{\mathbf{x}} - \mathbf{x}_{gt}\|^2. \tag{3}$$

**Dynamic Loss $\mathcal{L}_{dyn}$**    The dynamic loss measures how well the given sequence of states fits the considered dynamics. It is an unsupervised metric, formulated by

$$\mathcal{L}_{dyn}(\hat{\mathbf{x}}) = \|\Phi\left(\hat{\mathbf{x}}\right) - \hat{\mathbf{x}}\|^2 \tag{4}$$

where $\Phi$ is defined by

$$\begin{cases} \Phi(\mathbf{x})_0 & = \mathbf{x}_0 \\ \Phi(\mathbf{x})_i & = M_{pr}(\mathbf{x}_{i-1}; t_{i-1}, t_i), \ 1 \leq i \leq N - 1 \end{cases} \tag{5}$$

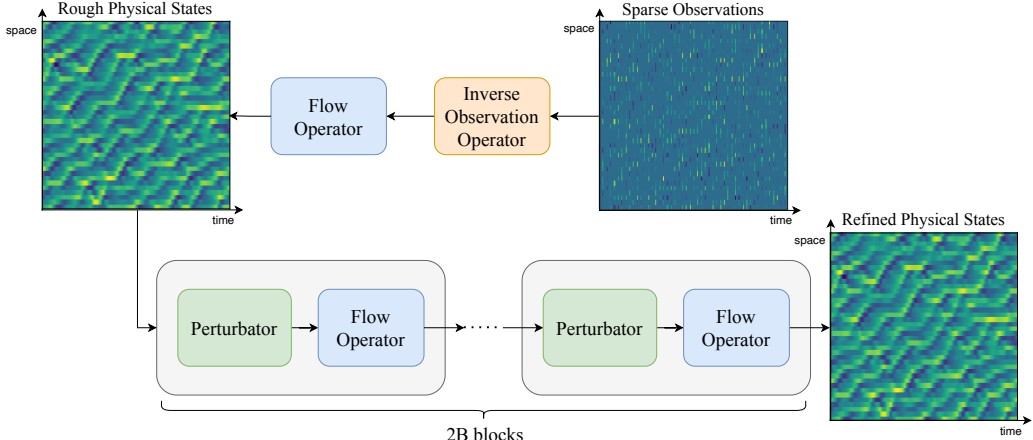

Figure 1: Proposed deep learning framework. The inverse observation operator (orange box) and the flow operator (blue box) aim to convert sparse observations to complete physical states and bring them closer to the prior dynamics, respectively, while the second row demonstrates the process of refining reconstructed physical states with the help of perturbators (green box) and flow operators.

$M_{pr}(\mathbf{x}_{i-1}; t_{i-1}, t_i)$ represents integrating $\mathbf{x}_{i-1}$ one step ahead from $t_{i-1}$ to $t_i$, using the prior dynamic model $\mathcal{M}_{pr}$. $\Phi(\mathbf{x})$ is also a sequence of states. The output at the $i$th time point, denoted by $\Phi(\mathbf{x})_i$, is obtained by integrating $\mathbf{x}_{i-1}$ one step forward. The RK45 integration scheme (Dormand & Prince, 1980) is adopted throughout our paper. The smaller the dynamic loss, the better the sequence of states fits the dynamics. If the sequence of states $\mathbf{x}$ perfectly matches the dynamic system, then $\Phi(\mathbf{x}) = \mathbf{x}$ holds (ignore the error of integration scheme), and the dynamic loss is zero.

## 3.2 HOW THE THREE OPERATORS WORK

**Inverse Observation Operator**    Our framework starts with the inverse observation operator $\mathcal{A}$, which aims to learn lower-dimensional observations and convert them into physical states (Frerix et al., 2021). With the help of the inverse observation operator, we are already able to construct the sequence of physical states, but with relatively low accuracy. Instead of seeking help from traditional 4D-Var algorithms to further improve accuracy (Frerix et al., 2021), we construct the so-called perturbator and the flow operator for such refining purposes.

**Flow Operator**    The main function of the flow operator is to reduce the dynamic loss of the reconstructed sequence of states and output a sequence that better conforms to the prior dynamics. Suppose that $\mathbf{x}$ is a reconstructed sequence of states, and we require that the dynamic loss after the operator is smaller than a given value $\mathcal{L}_0$, which is usually orders of magnitude smaller than the original dynamic loss. Denote the flow operator by $\mathcal{F}$, and the requirement can be formulated as

$$\mathcal{L}_{dyn}(\mathcal{F}(\mathbf{x})) \leq \mathcal{L}_0 \ll \mathcal{L}_{dyn}(\mathbf{x}) \tag{6}$$

We emphasize that although it is generally infeasible for neural networks to learn how to make arbitrary inputs satisfy prior dynamics, in our case, the input already conforms to the previous dynamics to some extent, which makes the goal achievable.

**Perturbator**    The role of the flow operator is to guarantee that the reconstructed sequence conforms to the dynamics, but this is insufficient for data assimilation, because there are infinitely many sequences of states that conform to the dynamics, and we need to find the one that also fits the observations well. On the other hand, thanks to the flow operator's ability to guarantee that the output is consistent with the prior dynamics, we can add a new neural network, called the perturbator, in front of the flow operator, which has more free space to utilize observational information by learning from ground truth labels. It can use the learned knowledge to revise the reconstructed sequence of states without considering the problem of conforming to the prior dynamics.

In this way, we **decouple** the two goals of conforming to observations and conforming to prior dynamics, and both the flow operator and the perturbator can perform their own duties. Specifically,

the perturbator uses what it has learned from the observations and labels to perturb the reconstructed states away from the original flow, and then the flow operator corrects the deviated flow to make it a new flow that conforms to the prior dynamics.

**How the Perturbator and the Flow Operator Work Together**    The perturbator and the flow operator always work in pairs. Given a sequence of states with relatively low dynamic loss $\mathcal{L}_{dyn}$ and relatively high reconstruction loss $\mathcal{L}_{rec}$, assuming that we have trained a flow operator $\mathcal{F}$, the function of the perturbator $\mathcal{P}$ can be formulated as

$$
\begin{aligned}
\mathcal{L}_{rec}(\mathcal{F}(\mathcal{P}(\mathbf{x}))) \leq \mathcal{L}_{rec}(\mathbf{x}) \\
\mathcal{L}_{dyn}(\mathcal{F}(\mathcal{P}(\mathbf{x}))) \sim \mathcal{L}_{dyn}(\mathbf{x})
\end{aligned}
\tag{7}
$$

The first formulation requires that the reconstruction loss is reduced, and the second requires that the dynamic loss should remain in the same order of magnitude. Ideally, as more pairs of perturbators and flow operators are applied to the input, the reconstruction loss will continuously decrease while maintaining the dynamic loss.

We emphasize that the output of the perturbator $\mathcal{P}(\mathbf{x})$ is likely to have both high reconstruction loss and high dynamic loss, but it does not matter because we only require that the losses after the flow operator $\mathcal{F}(\mathcal{P}(\mathbf{x}))$ are small.

### 3.3   Network Training

Our framework is trained using sequences of true states via supervised learning. Since the overall framework adopts a checkerboard style for refining the reconstructed states, we design a specific training procedure for all the three operators to work properly. We first use the reconstruction loss function to train a single inverse observation operator; then a flow operator is added after the inverse observation operator and trained with the dynamic loss. Afterwards, we gradually add pairs of perturbators and flow operators, and train and fine-tune them with a hybrid loss. More details are shown in Algorithm 1. We embed the unobserved regions with zeros to make the input observations $\mathbf{y}$ the same size as the output physical states $\mathbf{x}$.

**Stopping Condition**    In the case in which the prior dynamics are biased, we stipulate a stopping criterion for the training procedure in order to alleviate overfitting, which is formulated as follows:

$$
\mathcal{L}_{dyn}(\hat{\mathbf{x}}) \geq \mathbb{E}_{\mathbf{x}_{gt}}\left[\mathcal{L}_{dyn}(\mathbf{x}_{gt})\right]
\tag{8}
$$

where $\mathbb{E}_{\mathbf{x}_{gt}}\left[\mathcal{L}_{dyn}(\mathbf{x}_{gt})\right]$ is the expectation of the prior dynamic loss of ground truth states, which measures the difference between the prior dynamics and ground truth. It can be estimated in advance by averaging the dynamic loss over all samples in the training set. Each time the dynamic loss of the outputs $\mathcal{L}_{dyn}(\hat{\mathbf{x}})$ is calculated, we compare it with $\mathbb{E}_{\mathbf{x}_{gt}}\left[\mathcal{L}_{dyn}(\mathbf{x}_{gt})\right]$; if the inequality 8 is not satisfied, we end the current training phase and move on to the next one. Inequality 8 guarantees that the dynamic loss of the reconstructed states is no smaller than that of the true states. It is a *necessary but not sufficient* condition to avoid overfitting in terms of dynamics, and any reconstructed states with a smaller dynamic loss than true states indicate that more information than expected has been learned from the prior dynamics. Specifically, if the prior dynamics are accurate, then $\mathbb{E}_{\mathbf{x}_{gt}}\left[\mathcal{L}_{dyn}(\mathbf{x}_{gt})\right] = 0$, inequality 8 trivially holds, and we don't need to consider the stopping condition.

### 3.4   Network Structures for Operators

The network structures are based on the Residual U-Net structure (Ronneberger et al., 2015; Siddique et al., 2021) for all three operators, as shown in the Appendix. The hierarchical structure of U-Net allows it to capture both the large-scale and small-scale information of observation states and is ubiquitously used in the reconstruction domain (Guan et al., 2019; Siddique et al., 2021; Punn & Agarwal, 2022; Feng et al., 2020). Moreover, the residual structure helps alleviate the vanishing gradient problem (He et al., 2016), allowing U-Net models with deeper layers to be designed.

In addition, this paper also modifies the network according to the specificity of the data assimilation issue. Unlike images, of which the three dimensions are length, height, and channels, the inputs of data assimilation are sequences of observations (states), whose dimensions are time, space, and channels. The length and height of images are symmetric, while time and space are generally not. To tackle this difference, we carry out a dimensionality reduction by maxpooling only in the time dimension for the U-Net.

---

**Algorithm 1** Training procedure of our proposed framework (without stopping condition)

---

**Require:** : Ground Truth Dataset $\mathcal{D} = \{(\mathbf{y}, \mathbf{x}) \mid \text{observations } \mathbf{y} \in \mathbb{R}^{d \times N}, \text{states } \mathbf{x} \in \mathbb{R}^{d \times N}\}$
  Initialize the inverse observation operator $\mathcal{A}_\theta$, the flow operator $\mathcal{F}_\phi$, and the perturbator $\mathcal{P}_\psi$.
  Choose the number of blocks $2B + 2$, the hyperparameter $\lambda > 0$, and different epoch numbers $e_A, e_F, e_P$.
  **for** *epoch* from 1 to $e_A$ **do**       ▷ Phase 1: training the inverse observation operator.
    **for** each batch $(Y, X) \in \mathcal{D}$ **do**       ▷ $(Y, X)$ corresponds to the data batch of $(\mathbf{y}, \mathbf{x})$.
      $\hat{X} \leftarrow \mathcal{A}_\theta(Y), \mathcal{L}_\theta \leftarrow \mathcal{L}_{rec}(\hat{X})$
      Update $\theta$ with the Adam optimizer (Kingma & Ba, 2014).
    **end for**
  **end for**
  **for** *epoch* from 1 to $e_F$ **do**       ▷ Phase 2: training the flow operator.
    **for** each batch $(Y, X) \in \mathcal{D}$ **do**
      $\hat{X} \leftarrow \mathcal{F}_\phi(\mathcal{A}_\theta(Y)), \mathcal{L}_\phi \leftarrow \mathcal{L}_{dyn}(\hat{X})$
      Update $\phi$ with the Adam optimizer.       ▷ The inverse observation operator is fixed.
    **end for**
  **end for**
  **for** $k$ from 1 to $B$ **do**       ▷ Phases $3 \sim B + 2$: training the perturbator and fine-tuning.
    **for** *epoch* from 1 to $e_P$ **do**       ▷ *epoch* = 1: training; *epoch* > 1: fine-tuning
      **for** each batch $(Y, X) \in \mathcal{D}$ **do**
        $\hat{X} \leftarrow (\mathcal{F}_\phi \circ \mathcal{P}_\psi)^k \circ \mathcal{F}_\phi \circ \mathcal{A}_\theta(Y), \mathcal{L}_\psi \leftarrow \mathcal{L}_{rec}(\hat{X}) + \lambda \mathcal{L}_{dyn}(\hat{X})$
        Update $\psi$ with the Adam optimizer.       ▷ The flow operator is fixed.
      **end for**
    **end for**
    **for** *epoch* from 1 to $e_P$ **do**       ▷ *epoch* $\geq$ 1: fine-tuning
      **for** each batch $(Y, X) \in \mathcal{D}$ **do**
        $\hat{X} \leftarrow (\mathcal{F}_\phi \circ \mathcal{P}_\psi)^k \circ \mathcal{F}_\phi \circ \mathcal{A}_\theta(Y), \mathcal{L}_\phi \leftarrow \mathcal{L}_{rec}(\hat{X}) + \lambda \mathcal{L}_{dyn}(\hat{X})$
        Update $\phi$ with the Adam optimizer.       ▷ The perturbator is fixed.
      **end for**
    **end for**
  **end for**

---

As for the inverse observation operator, we add a skip layer from the input to the bottom hidden variables for better information transfer. As for the flow operator, we add a residual block from the input to the output to make the training process converge faster. More details can be found in the Appendix.

It is noted that our framework is not limited to the U-Net. More advanced networks, such as the work of Wang et al. (2020), can also be adopted in the future to better handle other more difficult chaotic systems.

## 4 NUMERICAL EXPERIMENTS

In this section, we design the numerical experiments to test the performance of our framework in different scenarios.

### 4.1 DESIGN OF EXPERIMENTS

**Dynamic Systems**    The first step is to select dynamic systems to test on. Lorenz systems are widely used for the evaluation of various data assimilation algorithms, owing to their chaotic properties and similarities with meteorological systems, especially in data-driven and machine learning case-studies (Lguensat et al., 2017; Raissi, 2018; Fablet et al., 2021b; Frerix et al., 2021). In our work, we choose the Lorenz-63 and Lorenz-96 systems for evaluation. The dimension of the Lorenz-63 system is three; the dimension of the Lorenz-96 system is modifiable, and we set it to 40.

**The Prior Dynamics**    The second step is to determine the prior dynamics, which is crucial for the training of the flow operator. We construct the prior dynamics by varying parameters in the ordinary differential equations (ODE).

The Lorenz-63 and Lorenz-96 dynamic systems are governed by the ordinary differential equations (ODE) expressed as Equation 9 and Equation 10, respectively (Lorenz, 1963; 1996).

$$\begin{cases} \frac{dX_1}{dt} & = \sigma \left( X_2 - X_1 \right) \\ \frac{dX_2}{dt} & = \rho X_1 - X_2 - X_1 X_3 \\ \frac{dX_3}{dt} & = X_1 X_2 - \beta X_3 \end{cases} \tag{9}$$

$$\frac{dX_i}{dt} = (x_{i+1} - x_{i-2}) x_{i-1} - x_i + F, \, i = 1, 2, ..., 40 \tag{10}$$

As for the ground truth dynamics, we assign $\sigma = 10$, $\rho = 28$, $\beta = 8/3$ for the Lorenz-63 system and $F = 8$ for the Lorenz-96 system. To construct the equations for the prior dynamics, we increase $\sigma$ by $\Delta \sigma$ in Equation 9 for the Lorenz-63 system and $F$ by $\Delta F$ in Equation 10 for the Lorenz-96 system.

**Dataset Construction**    Before constructing the dataset, parameters like $N$, $\Delta t$, $R$, $\Omega_{t_i}$ should be determined in advance. Refer to the Appendix for these experimental settings. We do a long integration over time using the ground truth dynamics and divide the time sequence into three sections. The first section is for building the training set, the second is for the validation set, and the last is for the test set. For each section, we continuously randomly intercept sequences of length $N$ as dataset samples. Also refer to the Appendix for detailed settings.

## 4.2 Unbiased Prior Dynamics

First, we consider the cases in which the prior dynamics are identical to the ground truth dynamics. We use a total of eight ($B = 3$) and ten blocks ($B = 4$) to train the deep learning framework for the Lorenz-63 and the Lorenz-96 systems, respectively, as shown in Figure 1.

We compare our framework with two baselines. The first is Fablet et al. (2021b), which is considered as the state-of-the-art. The second is a U-Net baseline, which shares the same structure of our framework, but follows a simple training procedure: (1) train the inverse observation operator and the flow operator in the front with the hybrid loss; (2) add one pair of the perturbator and the flow operator to the back, and then train the entire network with the hybrid loss; (3) repeat (2) until the number of blocks meets the setting of our models.

Apart from the reconstruction loss and the dynamic loss defined above, we evaluate the efficiency of our framework by test time. Test time refers to the time for the neural network to finish evaluation for one round on the test dataset. Shorter test time indicates higher computational efficiency.

The experimental results are demonstrated in Table 1. Our proposed method reduces the reconstruction error by 44% and 3% for the Lorenz-63 and the Lorenz-96 systems, respectively, and reduces the dynamic loss by two orders of magnitude, compared with Fablet et al. (2021b). In terms of test time, our proposed framework is four times and three times faster than Fablet et al. (2021b) in both dynamic systems. Our method also outperforms the U-Net baseline, in terms of both $\mathcal{L}_{rec}$ and $\mathcal{L}_{dyn}$, which proves the advantage of our training procedure.

We draw an example from the test dataset of the Lorenz-63 system to further illustrate how our architecture works, as shown in Figure 2. Focus on the blue oval box area in the upper right corner, where we find how the cooperation of the perturbator and the flow operator succeeds in correcting the rough reconstructed sequence of states. In the beginning, the physical states are clearly inconsistent with the ground truth in the upper right region. After a perturbator is added, disturbances with high frequencies are involved in the misaligned area to distinguish them from well-constructed areas. If a flow operator is further added behind the perturbator, the high-frequency noise disappears, and the inconsistency is corrected to some extent. Such a perturbation-refinement step is repeated and the misalignment becomes smaller with repetition.

| Dynamic System | Model | # of Blocks | $\mathcal{L}_{rec}$ | $\mathcal{L}_{dyn}$ | AVG (s) | SD (s) |
|---|---|---|---|---|---|---|
| Lorenz-63 | Our work | 2 | 1.28 | 1.10e-3 | 3.1 | 0.65 |
| | Our work | 4 | 0.923 | 3.82e-4 | 4.5 | 0.48 |
| | Our work | 6 | 0.799 | 2.08e-4 | 5.8 | 0.63 |
| | Our work | 8 | **0.751** | **1.52e-4** | 6.4 | 0.40 |
| | U-Net baseline | 8 | 0.849 | 3.99e-4 | 7.2 | 0.56 |
| | Fablet et al. (2021b) | 20 | 1.34 | >1.00e-2 | 27.1 | 3.80 |
| Lorenz-96 | Our work | 2 | 0.571 | 1.40e-3 | 7.8 | 0.60 |
| | Our work | 4 | 0.450 | 5.09e-4 | 9.8 | 0.97 |
| | Our work | 6 | 0.397 | 3.14e-4 | 11.5 | 0.93 |
| | Our work | 8 | 0.373 | 2.66e-4 | 13.5 | 0.77 |
| | Our work | 10 | **0.367** | **2.36e-4** | 15.8 | 0.72 |
| | U-Net baseline | 8 | 0.599 | 2.24e-3 | 12.8 | 0.88 |
| | Fablet et al. (2021b) | 20 | 0.38 | >1.00e-2 | 47.3 | 1.46 |

Table 1: Results of comparative experiments. We report the performance of the proposed framework and two baselines for the Lorenz-63 and Lorenz-96 systems. The results of Fablet et al. (2021b) draw directly from the original paper (Fablet et al., 2021b). Bold values refer to the best score. All of the models are tested on a test set with 2000 samples for Lorenz-63 and 3000 samples for Lorenz-96. The batch size is set to eight, and all the evaluations are performed on one TITAN RTX GPU. Test time may vary across experiments, and we provide the average (AVG) and standard deviation (SD) of test time from 10 independent tests.

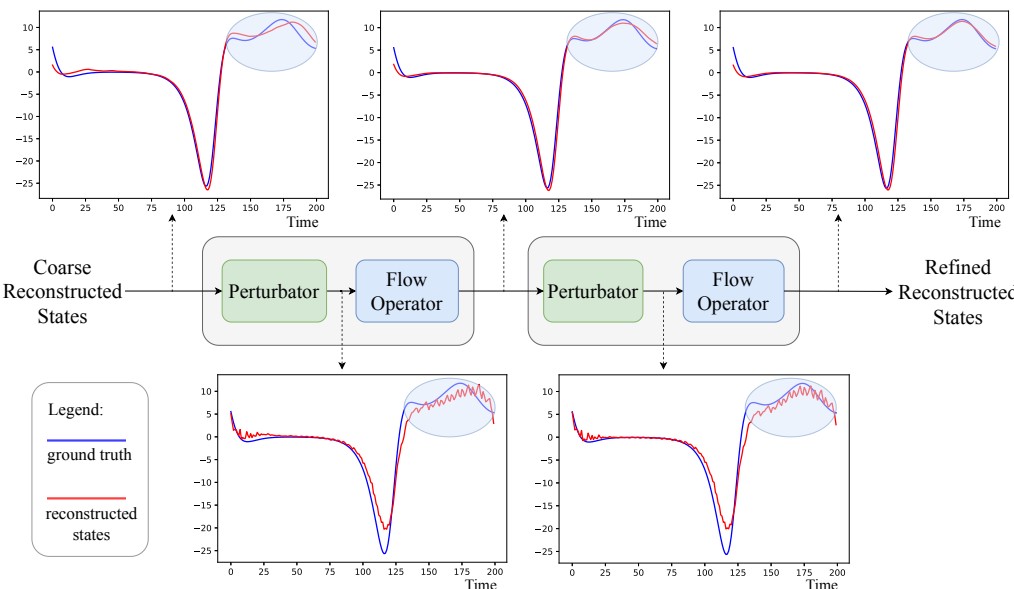

Figure 2: An intuitive presentation of how the refining structure works. A sample is randomly drawn from the test dataset of the Lorenz-63 system for demonstration. The rough reconstructed states are generated by the first two operators (the inverse observation operator and the flow operator). We use two pairs of perturbators and flow operators to refine the input, and extract the intermediate results generated by each block to demonstrate the detailed procedure.

## 4.3 BIASED PRIOR DYNAMICS

Next, we consider the cases in which the prior dynamics are biased. Specifically, we select $\Delta\sigma$ from the set {0.25, 0.50, 0.75, 1.00, 1.50, 2.00, 2.50, 3.00, 3.50, 4.00, 4.50, 5.00} for the Lorenz-63 system, select $\Delta F$ from the set {0.50, 1.00, 1.50, 2.00, 2.50, 3.00, 3.50, 4.00, 4.50, 5.00} for the Lorenz-96 system, and conduct numerical experiments accordingly.

We report our results in Figure 3. Take the Lorenz-63 system as an example. As illustrated in the upper-right figure, when $\Delta\sigma$ is smaller than 1.0, the increase in $\mathcal{L}_{rec}$ is smaller than 20%, which shows that our framework is robust to small biases. When $\Delta\sigma$ is smaller than 2.00, the $\mathcal{L}_{rec}$ increases as $\Delta\sigma$ increases, whereas if $\Delta\sigma$ is larger than 2.00, the $\mathcal{L}_{rec}$ fluctuates without monotonic changes. Such a phenomenon can be explained by looking into how our network structure works: When $\Delta\sigma$ is small, the difference between the prior dynamics and the ground truth is small, and the information of the prior dynamics can be successfully exploited by the flow operator. Therefore, as $\Delta\sigma$ gets larger, useful information provided by the prior dynamics decreases and the performance of our framework worsens. However, if $\Delta\sigma$ is too large, it is no longer necessary for the flow operator to learn from the bad prior dynamics; it is cheaper to learn directly from ground truth labels. Hence, in this case, the value of $\Delta\sigma$ does not affect the performance of the neural network. The results shown in the upper-left figure can further confirm our explanation. When $\Delta\sigma < 2$, the difference between $\mathcal{L}_{dyn}(\hat{\mathbf{x}})$ and $\mathcal{L}_{dyn}(\mathbf{x}_{gt})$ is close to zero, which is consistent with our stopping criterion (Inequality 8), so the network works as expected. However, if $\Delta\sigma > 2$, the bias of prior dynamics is too large for the flow operator to learn, and solely learning from the ground truth is sufficient for the neural network to generate states whose dynamic loss is smaller than $\mathcal{L}_{dyn}(\mathbf{x}_{gt})$. The findings above also hold for the Lorenz-96 system, as shown in the lower figures. The difference is that the threshold changes from 2.0 to around 2.5. They provide further evidence that our network learns features from prior dynamics rather than simply remembering ground-truth labels, and that our framework works properly despite biases in prior dynamics.

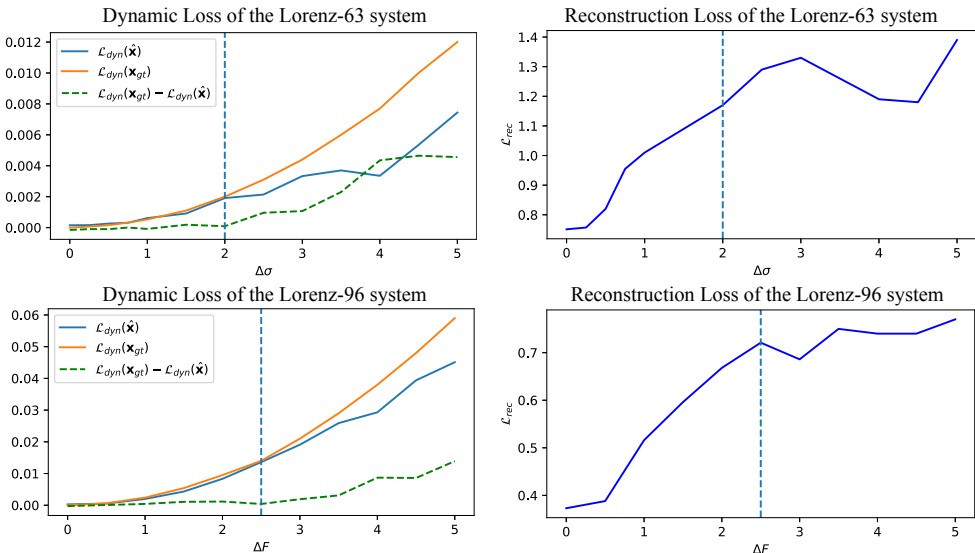

Figure 3: The left panels show the dynamic loss of reconstructed states (blue line), the biases of prior dynamics (orange line), and their differences (green dotted line) against $\Delta\sigma$ or $\Delta F$. The right panels show the reconstruction loss ($\mathcal{L}_{rec}$) in the test set against $\Delta\sigma$ or $\Delta F$. The upper and lower panels represent the results of the Lorenz-63 and Lorenz-96 systems, respectively.

## 5 CONCLUSIONS

We have introduced a deep learning framework for data assimilation. The key novelty of our work is that we develop the flow operator to learn from the prior dynamics and continuously refine the reconstructed sequence of states. The results of numerical experiments on the Lorenz-63 and Lorenz-96 systems support the relevance of our framework. In terms of assimilation accuracy, our framework has reduced the dynamic loss by two orders of magnitude, and at least maintained the reconstruction accuracy, compared with the state-of-the-art method. In terms of computational efficiency, our framework has been found to be four times and three times faster than the state-of-the-art method on the Lorenz-63 and Lorenz-96 systems, respectively. It has also been found to be well-adapted to biases in the prior dynamics.

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

## A    DETAILED NETWORK STRUCTURE FOR OPERATORS

We take the three-layer U-Net as an example to show the detailed network structures for the inverse observation operator (Figure 4) and the perturbator (Figure 5). The structure of the flow operator is almost the same as that of the perturbator, except that a skip layer is added from the input to the output.

## B    EXPERIMENTAL SETTINGS

### B.1    DYNAMIC SYSTEMS SETUP

In the Lorenz-63 system, $t = 0.01$, $N = 200$, $R = 8$, $Var[\epsilon(t)] = 2.0$, $\Omega_t = \{1\}$. This means that the time step for integration is 0.01, and the number of time points in the assimilation window is 200. Only the first component of the physical states is observed. The observations are sampled every eight steps, and we also add a Gaussian noise with a variance of 2.0 to them.

In the Lorenz-96 system, $t = 0.05$, $N = 200$, $R = 4$, $Var[\epsilon(t)] = 2.0$, $\Omega_t = \{$sample 20 random integers between 1 and 40 for each $t\}$. This means that the time step for integration is 0.05, and the number of time points in the assimilation window is 200. The observations are sampled every four steps, and each time we observe 20 random components of the physical states. Gaussian noise with a variance of 2.0 is also added to the observations.

### B.2    DATASET INFORMATION

In the Lorenz-63 system, the training data comprises 10000 sequences sampled from time series with 12000 steps; the validation data comprises 2000 sequences from time series with 5000 steps and the test data comprises with 3000 sequences from 10000 time steps. In the Lorenz-96 system, the training data comprises 2000 sequences sampled from time series with 7500 steps; the validation

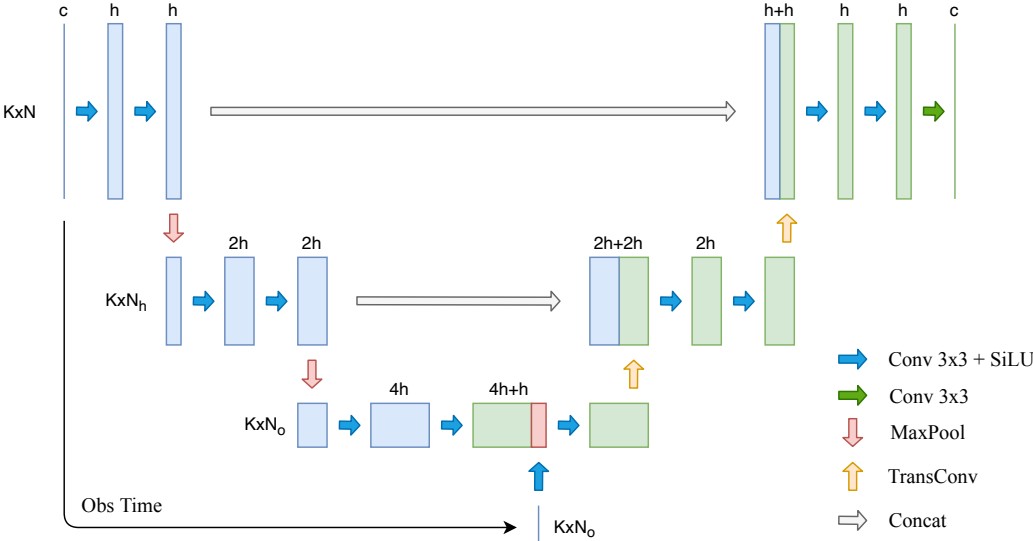

Figure 4: An example of the 3-layer U-Net architecture of the inverse observation operator. We denote $K$ the spatial dimension of the physical states, $c$ the number of channels (the total dimension of the system is $d = c \times K$), $N$ the total sequence length, $h$ the hidden dimension, $N_o$ the sequence length with valid observations, $N_h$ the intermediate hidden sequence length. The input of the inverse observation operator is observations. Before input, the sparse observations are padded to the same size as the output sequence of physical states by padding unobserved regions with zeros.

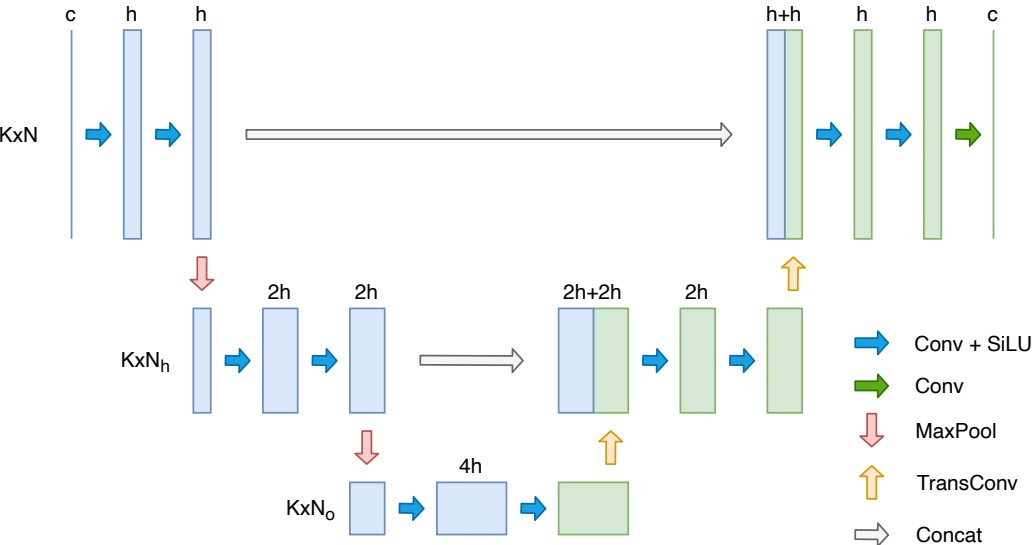

Figure 5: An example of the 3-layer U-Net architecture of the perturbator. We let readers refer to Figure 4 for details. Both the input and output of the perturbator are a sequence of physical states.

data comprises 300 sequences from time series with 2500 steps and the test data comprises with 2000 sequences from 10000 time steps. Settings for the training set and validation set correspond to those in Fablet et al. (2021b), and we increase the size of test data to increase the credibility of experimental results.

### B.3 NETWORK SETTINGS

In the Lorenz-63 experiments, the input is reshaped as $c \times K \times N$, where $c = 3$, $K = 1$, and $N = 200$. A four-layer U-Net is adopted, with hidden dimension $h = 32$. In the Lorenz-96 experiments, the input is reshaped as $c \times K \times N$, where $c = 1$, $K = 40$, and $N = 200$. A three-layer U-Net is adopted, with hidden dimension $h = 32$. In both experiments, $\lambda = 100$, $e_A = 20$, $e_F = 200$, $e_P = 20$ for training according to Algorithm 1.

## C DETAILED RESULTS OF TRAINING

### C.1 TRAINING COST

The training cost of our framework is affordable on one TITAN RTX GPU. The training speed is around three times that of Fablet et al. (2021b). Details are demonstrated in Table 2 and Table 3 for the Lorenz-63 and Lorenz-96 systems, respectively.

| Model | Phase | Time per epoch | # of epochs | Phase time |
|---|---|---|---|---|
| Total time | | | | |
| Fablet et al. | Phase 1 (10 blocks) | 1 m 38 s | 20 | 32 m 40 s |
| (2021b) | Phase 2 (10 blocks) | 1 m 45 s | 100 | 175 m |
| | Total time | | | 207 m 40 s |
| Our work | Phase 1 | 7 s | 20 | 2 m 20 s |
| | Phase 2 | 10 s | 200 | 33 m 20 s |
| | Phase 3 (4 blocks) | 24 s | 20 | 8 m |
| | Phase 4 (6 blocks) | 36 s | 20 | 12 m |
| | Phase 5 (8 blocks) | 44 s | 20 | 14 m 40 s |
| | Total time | | | 70 m 20 s |

Table 2: Training time for the Lorenz-63 system. If the validation loss does not continue to decrease, we consider the network to converge and end the current phase. The training procedure of Fablet et al. (2021b) is divided into two phases: the first phase is trained with 10 blocks and the second phase is trained with 20 blocks. We should at least allocate 20 epochs for the first phase and 100 epochs for the second phase to make the training loss converge well. We trained our framework up to eight blocks, strictly following the training procedure of Algorithm 1. The phases of our work correspond to those defined in Algorithm 1. Both models are trained on a TITAN RTX GPU with batch size set to 8. The statistics of time per epoch, number of epochs for each phase, phase time and total time are shown in the table.

| Model | Phase | Time per epoch | # of epochs | Time |
|---|---|---|---|---|
| Fablet et al. | Phase 1 (10 blocks) | 2 m 34 s | 20 | 51 m 20 s |
| (2021b) | Phase 2 (10 blocks) | 4 m 45 s | 100 | 475 m |
| | Total time | | | 526 m 20 s |
| Our work | Phase 1 | 9 s | 20 | 3 m |
| | Phase 2 | 33 s | 200 | 110 m |
| | Phase 3 (4 blocks) | 33 s | 20 | 11 m |
| | Phase 4 (6 blocks) | 48 s | 20 | 16 m |
| | Phase 5 (8 blocks) | 63 s | 20 | 21 m |
| | Total time | | | 161 m |

Table 3: Training time for the Lorenz-96 system. We let readers refer to Table 2 for details.

### C.2 LOSS CHANGE DURING TRAINING

The variations of the reconstruction loss and the dynamic loss against the number of epochs for the Lorenz-63 and Lorenz-96 systems are reported in Figure 6 and Figure 7, respectively. In the first phase, the inverse observation operator learns a mapping from the observational space to the physical states. Since a shift between two different spaces takes place in this period, we see a steep decrease

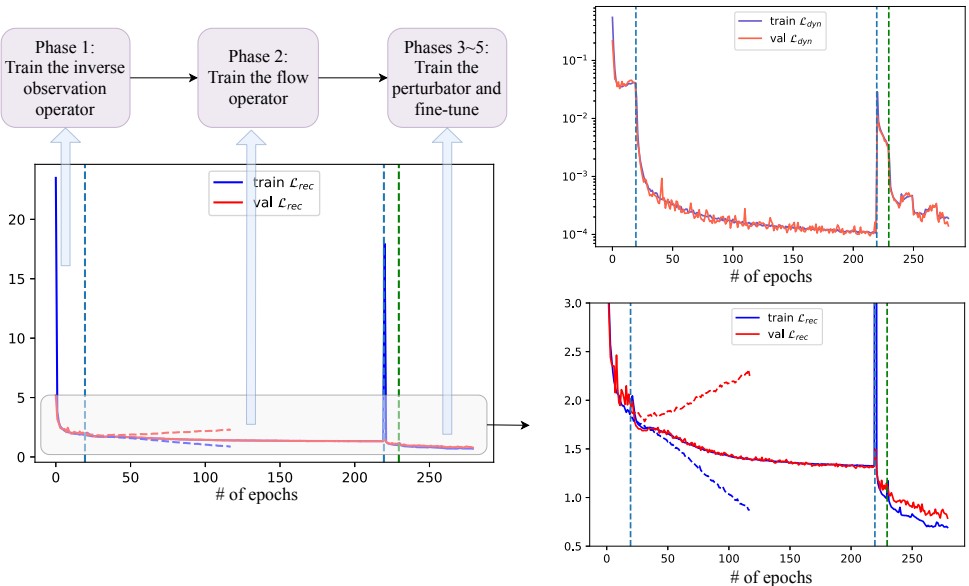

Figure 6: The variation of the reconstruction loss $\mathcal{L}_{rec}$ and the dynamic loss $\mathcal{L}_{dyn}$ for training and validation with the number of epochs for the Lorenz-63 system. Two blue vertical dashed lines divide the training procedure into three periods: training the inverse observation operator, training the flow operator, and phases 3∼5. The green vertical further divide the phases 3∼5 to two periods: training the perturbator and fine-tuning. The left panel demonstrates the changes in the reconstruction loss during training, the details of which are enlarged and shown in the bottom right panel. The red and blue dashed lines in the left panel and the bottom right panel show how continuing to train the inverse observation operator without adding new operators would result in a change in the reconstructed loss. The top right panel demonstrates the variation of dynamic loss during training.

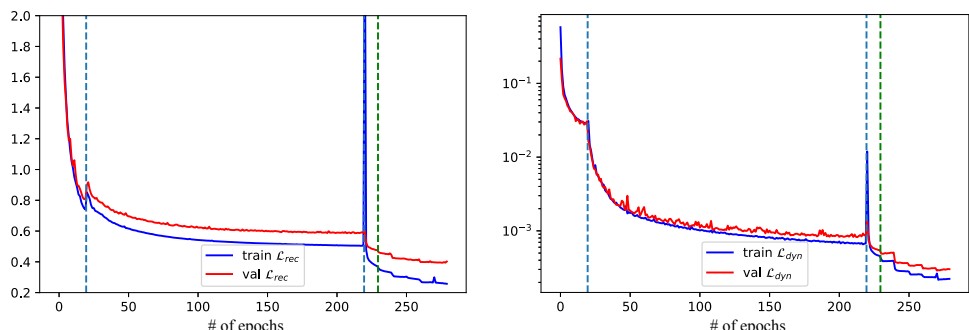

Figure 7: The variation of the reconstruction loss $\mathcal{L}_{rec}$ and the dynamic loss $\mathcal{L}_{dyn}$ with the number of epochs for the Lorenz-96 system. The left panel demonstrates the changes in the reconstruction loss during training, while the right panel demonstrates the variation of the dynamic loss.

of loss within the first 20 epochs. After the training of the inverse observation operator is finished, we start the training of the flow operator to correct the flow dynamics of the reconstructed states. The training of the flow operator is relatively slow because an unsupervised loss is employed in this procedure. We claim that the flow operator is indispensable in further decreasing the reconstruction loss and dynamic loss, for the reason that without the flow operator, the inverse observation operator would quickly lead to overfitting, in other words, the difference between the training loss and validation loss will diverge, as shown in the red and blue dashed lines in the figure. There is a sudden

increase in the loss function after the second blue vertical line because a perturbator with untrained parameters enters our framework. During the fine-tuning procedure, the reconstruction loss drops steadily while the dynamic loss fluctuates. This is not surprising considering that the fine-tuning procedure alternates between the perturbators and the flow operators.

Reconstructed examples and associated absolute errors

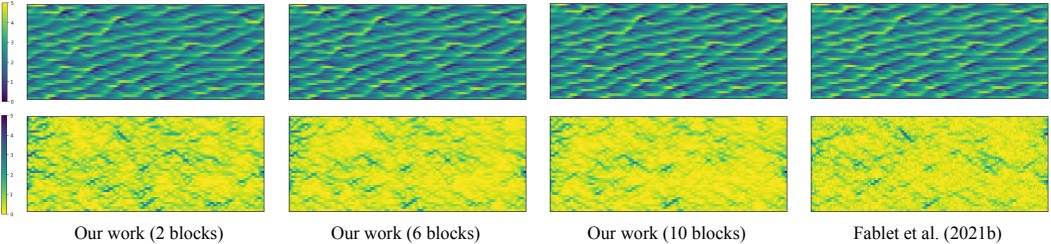

Our work (2 blocks)   Our work (6 blocks)   Our work (10 blocks)   Fablet et al. (2021b)

Figure 8: The sequences of reconstructed states and their corresponding absolute error maps of the Lorenz-96 system, with the horizontal axis being the time axis and the vertical axis being the spatial axis. All the displayed figures in the same row share the same colormap.

## D    SAMPLE STUDY OF THE LORENZ-96 SYSTEM

Figure 8 demonstrates the physical states reconstructed by different neural networks for the Lorenz-96 system and their associated absolute error maps. Visually, the dark regions in the error map becomes smaller and lighter as the number of block grows, which confirms the effects of the collaboration of the perturbators and the flow operators in this example.

## E    FURTHER EXPERIMENTS

### E.1    COMPARATIVE STUDY ON THE BIASED PRIOR DYNAMICS

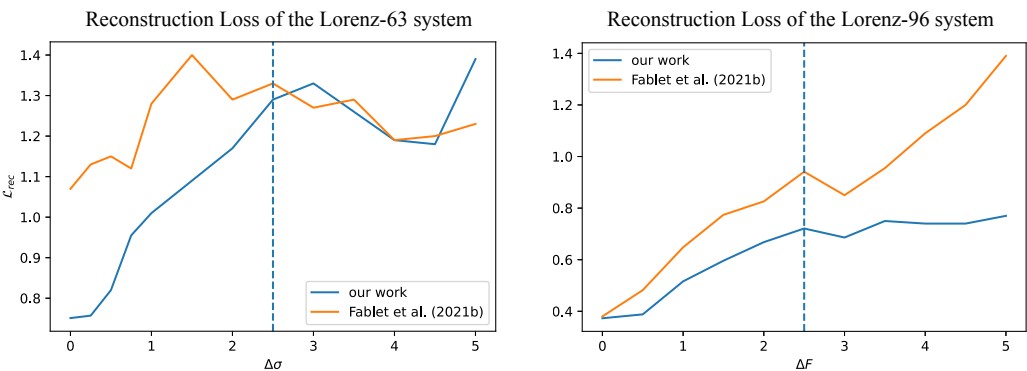

Figure 9: Results of our work and Fablet et al. (2021b) under biased prior dynamics. The left panel shows the reconstruction loss ($\mathcal{L}_{rec}$) against $\Delta\sigma$ for the Lorenz-63 system; the right panel shows the reconstruction loss against $\Delta F$ for the Lorenz-96 system.

Here, we compare the performance of our work with Fablet et al. (2021b) under biased prior dynamics. Since Fablet et al. (2021b) did not realize their algorithms under this experimental setting, we implement their algorithms based on our best understanding of their work. The results are shown in Figure 9. It is worth noting that the results of the case in which bias equals zero are close to those presented in their paper, which confirms the credibility of our experiments.

As for the Lorenz-63 system, the reconstruction loss of their model is larger than that of our model as the prior bias is smaller than 2.5, and converges to around 1.20∼1.35 as the prior bias gets larger

than 2.5, which is similar to the performance of our model. As for the Lorenz-96 system, the reconstruction loss of their model seems to diverge as the prior bias gets larger, while the reconstruction loss of our model converges to around 0.7. These comparative experiments show that our proposed model is more adaptive to the prior bias than the baseline model. The difference in performance of their model under different dynamical systems can be attributed to the use of two different neural networks in the two cases.

### E.2 ABLATION STUDY

We add two ablation experiments to show that 1) the refining process by perturbators and flow operators greatly contributes to reducing the reconstruction error of the states generated by the inverse observation operator; 2) when the experimental setting for observations is changed, only the inverse observation operator needs retraining, and the network parameters for refining process can remain unchanged.

**Experiment 1** We remove all the perturbators and flow operators, and test the performance of one single inverse observation operator. It has been found that the checkerboard-style refining process can further reduce the reconstruction error by at least 50%, compared with one single inverse observation operator. The detailed results are shown in Table 4.

| Dynamic system | Network structure | $\mathcal{L}_{rec}$ | $\mathcal{L}_{dyn}$ |
|---|---|---|---|
| Lorenz-63 | InvObsOp only | 1.95 | >1.00e-2 |
| | Three operators (our work) | **0.751** | **1.52e-4** |
| Lorenz-96 | InvObsOp only | 0.763 | >1.00e-2 |
| | Three operators (our work) | **0.367** | **2.36e-4** |

Table 4: Results of the ablation study with the perturbators and the flow operators removed. "InvObsOp" is short for "inverse observation operator". The first row for each dynamic system corresponds to the network with only the inverse observation operator; the second row corresponds to our entire network, with eight and ten blocks, respectively.

**Experiment 2** Based on the Lorenz-96 system, we design two different settings for observations. In the first setting, the observations are sampled every four steps for a random half of the components; in the second setting, the observations are sampled every four steps for the components whose indexes are even. We conduct the experiments by the following steps.

Step 1: Train all three operators in the first experimental setting.

Step 2: Only train the inverse observation operator in the second experimental setting.

Step 3: Concatenate the inverse observation operator trained in Step 2 with the perturbator and the flow operator trained in Step 1 (Without further training or tuning).

Step 4: Test the concatenated network in the second experimental setting.

The results are reported in Table 5. Meanwhile, we also report the results of another experiment, in which the observation settings are switched, and the same experiment procedure is conducted again with the above steps. It can be seen that when the perturbator and the flow operator are transferred to another observation setting, they still work well, despite a slight increase in the reconstruction error (smaller than 10%). It also shows that the function of the observation operator and that of the perturbator and the flow operator can be decomposed well.

### E.3 EXPERIMENTS ON A NON-LORENZ SYSTEM

In order to evaluate our framework's applicability to other chaotic systems, we test it on Vissio-Lucarini20 (Vissio & Lucarini, 2020) system. Vissio-Lucarini20 is coupled system intended to represent a minimal model of the earth's atmosphere, which is composed of two types of variables, namely, kinetic variables and thermodynamical variables. Since both of the variables are coupled, this system is more complicated than the Lorenz systems. The corresponding ordinary differential

| Observation setting (for test) | Observation setting (for training) | $\mathcal{L}_{rec}$ | $\mathcal{L}_{dyn}$ |
|---|---|---|---|
| Random | Random | 0.367 | 2.36e-4 |
| | Even | 0.402 | 3.61e-4 |
| Even | Random | 0.304 | 3.61e-4 |
| | Even | 0.291 | 2.56e-4 |

Table 5: Results of the ablation study with the refining network (the perturbators and the flow operators) switched, on the Lorenz-96 system. The first column indicates the observation setting corresponding to the test results; the second column indicates which observation setting the perturbators and the flow operators are trained in. "Random" corresponds to the first setting, and "Even" corresponds to the second one.

equation is shown in Equation 11, where $k = 1, 2, ..., N$.

$$\begin{cases} \frac{dX_k}{dt} & = X_{k-1}(X_{k+1} - X_{k-2}) - \alpha\theta_k - \gamma X_k + F \\ \frac{d\theta_k}{dt} & = X_{k+1}\theta_{k+2} - X_{k-1}\theta_{k-2} + \alpha X_k - \gamma\theta_k + G \end{cases} \tag{11}$$

**Experimental settings** The numbers of both kinetic variables and thermodynamical variables are set to 40. $\alpha = \gamma = 1.0, F = 10, G = 0$. The observations are sampled every four steps for a random half of both kinds of variables. The variance of Gaussian noise for observations is 2.0. The prior dynamics are set to be unbiased.

**Network design** In our proposed method, we adopt a three-layer U-Net similar to that in the Lorenz-96 system. The difference is that we change the channel number from one to two, in order to fit the characteristics of the system with two types of variables. As for the baseline, Fablet et al. (2021b) does not implement their neural network for Vissio-Lucarini20 system, and we do our best to realize their algorithm.

**Experimental results** Similar to the results in the Lorenz-96 system, our network outperforms Fablet et al. (2021b) in terms of both the dynamic loss and test time. We report the details in Table 6.

| Dynamic System | Model | # of Blocks | $\mathcal{L}_{rec}$ | $\mathcal{L}_{dyn}$ | AVG (s) | SD (s) |
|---|---|---|---|---|---|---|
| Vissio-Lucarini20 | Our work | 10 | **0.352** | **3.57e-4** | **23.6** | **1.5** |
| | Fablet et al. (2021) | 20 | 0.358 | 3.25e-2 | 64.2 | 3.4 |

Table 6: Results of comparative experiments. We report the reconstruction and dynamical loss of the proposed framework for the Vissio-Lucarini20 system. Bold values refer to the best score. Both of the models are tested on an independent test set with 2000 samples. The batch size is set to eight, and all the evaluations are performed on one TITAN RTX GPU. Test time may vary across experiments, and we provide the average and standard deviation of results from 10 independent tests.

