# OpenReview forum: "Learning Dynamical Characteristics with Neural Operators for Data Assimilation"
_ICLR.cc/2023/Conference — Submitted to ICLR 2023_

### Official Review · Reviewer_BfUY · 2022-10-22

**Confidence:** 4
**Correctness:** 4
**Technical Novelty And Significance:** 3
**Empirical Novelty And Significance:** 3
**Recommendation:** 8

**Clarity, Quality, Novelty And Reproducibility:**

The paper is well written and on the most part easy to read. The novelty seems to be interesting, but a clearer explanation on the differences with Fablet et al. 2021 would be important to see in the contributions section. I did not see a code link, did I miss it?


**Strength And Weaknesses:**

Strenghs:
	- the problem tackled is of importance for a large community
	- the proposed method is end-to-end and able to provide a solution from both observation and prior dynamics
	- the results are promising, reducing considerably the loss dynamics and the computational time from previous study
	- the pipeline is efficient and the different U-net blocks are assembled in an intelligent way
	- the prior dynamic bias version seems to indeed work, which is one of the main novelties and assets.

Weaknesses:
	- The technique seems to be in its early stages and it seems as if tuning needs to be performed for every new dynamical system and setting. First, the different training phases are difficult to follow and makes me wondering if the pipeline is robust enough. Also, please explain how was chosen to use a 4-layer U-Net for the Lorenz-63 and only a 3-layer U-Net for the Lorenz-96.
	- Network structure: it is not clear which losses are minimized at the different phases, in particular, the perturbator+flow operator is not clear. From eq. 11, it looks as if the dynamics and the data fidelity losses were applied on the same current state x_hat, while my understanding was that L_rec was calculated on the perturbator's output while L_dyn was calculated on the flow operator's output. If the hybrid loss is calculated on the final output of the 2 blocks, I don't se how we can enforce the decoupling of the two goals as stated.
	- It is difficult to understand the size of the input data and latent data. In particular, please give axis labels on Figure 1 images, as it first looks like a 2D spatial problem, while later it is explained that the input is of size time and location. Please also clarify what does T represent in Figure 2: I guess the time dimension, and in this case where is the location dimension?
	- Please explain best the following sentence: 'For the case in which the prior dynamics are unbiased': in a real setting, how can we know if the prior dynamics are biased or not?
	- 'espilon(t) represents the white Gaussian noise.' --> isn't it reductive?
	- Why stopping at 8 and 10 blocks for the experiments, while it looks as if the results are always improving with more blocks?

Typos:
Specifically, The perturbator

The perturbator uses the observations and labels it has learned to perturb the reconstructed states to make it deviate from the original flow --> please rephrase

Figure 3: the color do not match Figure 1, as here the colors are also linked to training/no training. Maybe use another sign to indicate training/no training, as a red line surrounding the box.

**Summary Of The Paper:**


The paper proposes a deep learning framework for data assimilation technique for physics and earth science, such as meteorological dynamical processes. The framework is based on an estimation of a coarse state of the system from noisy and sparse observations, associated to a refinement using dynamical priors. Do do so, different U-net based architectures are developed: an observation operator, a flow operator, and a perturbator. Numerical experiments were performed on Lorenz-63 and Lorenz-96 one-dimentional dynamical systems, having 3 state variables. Both models were also tested with biased versions of the prior dynamics, which reproduces more accurately realistic situations.

**Summary Of The Review:**

This paper is interesting and proposes a new end-to-end deep learning framework for a difficult yet important application. In particular, the authors tackled the prior dynamics bias problem. The method seems to be in its early stages, but I think it would be beneficial for the community. I have to add that while I have some knowledge in the field, I am not a data assimilation expert.

---

> ### Author Response · Authors · 2022-11-18
> **Response to Reviewer BfUY (1/3)**
>
> We sincerely appreciate your careful and thoughtful comments and time, especially your approval for our proposed framework, which really means a lot to us. We try to explain your concerns point by point.
>
> **Q1: Robustness and applicability of the pipeline**
>
> **A1:**
>
> *Robustness:* From all the experiments we have done so far, the pipeline of our framework is shown to be robust.
>
> First, the training procedure is stable. In most of the experiments we have done, the training loss decreases steadily during the whole process, as shown in the appendix.
>
> Second, the framework is robust to the selection of hyperparameters. For example, in the formulation of hybrid loss, the selection of the parameter $\lambda$ affects the final accuracy of the framework, but regardless of the value of $\lambda$, as long as it is no larger than 300, the training loss does not diverge.
>
> Third, the framework is also robust to the selection of neural networks. For example, if we change the observation operator from the modified U-Net to an ordinary U-Net, the framework still works, although the accuracy is somewhat hindered.
>
> The reason why our pipeline is robust is that 1) every operator in our framework is physically explainable, and the behavior of each block can be tracked; 2) the training process matches the functions of our designed operators well.
>
> *Applicability:* When the dynamic system is completely changed, we have to train the entire framework from scratch. How to design a data assimilation neural network architecture that can be transferred across dynamic systems is a huge challenge, and we will deal with them in our future work.
>
> Nevertheless, we claim that when the experimental setting for observations is changed, only the observation operator needs retraining, and the network parameters for the refining process can remain unchanged. We design the following experiment to support this statement.
>
> *Experiment:* Taking the Lorenz-96 system as an example, we design two different settings for observations. For the first setting, the observations are only a random selection of half of the components in the Lorenz system and we sample the observations every four steps; For the second setting, the observations are sampled every four steps for the components whose indexes are even. We conduct the experiments according to the following steps.
>
> Step 1: Train the framework with all three operators under the first experimental setting. We use it as a well-trained reference model.
>
> Step 2: Only train the observation operator under the second experimental setting. We use it to simulate the case where only the observation operator is trained after the observation changes
>
> Step 3: Concatenate the observation operator trained in Step 2 with the perturbator and the flow operator trained in Step 1 without further training.
>
> Step 4: Test the combined network under the second experimental setting.
>
> The results are reported in the following table. Meanwhile, we also report the results of another experiment, in which the observation settings are switched, and the same experiment procedure is conducted again with the above steps. It can be seen that when the perturbator and the flow operator are transferred to another observation setting, they still work well, despite a slight increase in the reconstruction error (smaller than 10%). It also shows that the function of the observation operator and that of the perturbator and the flow operator can be decomposed well.
>
> | Observation setting | In which observation setting the perturbator and flow operator are trained | R-score | LossDyn |
> | ------------------- | ------------------------------------------------------------ | ------- | ------- |
> | Random              | Random                                                       | 0.367   | 2.36e-4 |
> |                     | Even                                                         | 0.402   | 3.61e-4 |
> | Even                | Random                                                       | 0.304   | 3.61e-4 |
> |                     | Even                                                         | 0.291   | 2.56e-4 |
>
> **Q2: Number of layers chosen**
>
> **A2:** The U-Net we choose for the Lorenz-63 system is one layer deeper than that for the Lorenz-96 system, because the observations are sampled every 8 steps for Lorenz-63 and every 4 steps for Lorenz-96, and we hope to better extract the sparser observational information for the Lorenz-63 system with a deeper network.

---

> ### Author Response · Authors · 2022-11-18
> **Response to Reviewer BfUY (2/3)**
>
> **Q3: Loss at different phases**
>
> **A3:** When tuning the parameters of the perturbator, the hybrid loss is also calculated on the real output of the 2 blocks (the perturbator and flow operator pair). As stated in the paper, the role of the perturbator is to use the learned knowledge to perturb the reconstructed states to help the flow operator generate better physical states. The perturbator and the flow operator work together, and we cannot separate the perturbator apart.
>
> Decoupling the two goals of conforming to observations and conforming to prior dynamics is our final expectation. Our current framework does not guarantee a theoretical complete decoupling because the perturbator is trained in an indirect manner. In future work, we will improve our framework to better enforce decoupling.
>
> **Q4: The size of the input and latent data & What T represents in Figure 2**
>
> **A4:** $T$ represents the time dimension. As shown in Figure 2 in the old version, the size of the input is $c\times K\times T$, and the size of latent data is $c\times K\times T_h$. Among them, $c$ represents the channels. For example, in real-world scenarios, if we want to model both the temperature and pressure, $c$ is set to two because there are two kinds of variables. $K$ represents spatial dimension. For example, if we want to model the temperature of ten grid points on a line, $K$ should be set to 10. $T$ represents the time dimension. For example, if 200 time points as a time window are considered for data assimilation, then $T$ should be set to 200. $T_h$ is the new time dimension after reduction, for example, we can set $T_h = T/2$.
>
> The Lorenz-63 system can be regarded as modeling three different kinds of variables at one single point. Therefore, the channel number $c$ is set to 3, and the spatial dimension $K$ is set to one. The input size is $3\times 1\times 200$. The Lorenz-96 system, on the other hand, can be regarded as modeling one variable at different grid points on a circle. In our experimental setting, the number of grid points is set to 40. Hence, the channel number $c$ equals one, while the spatial dimension $K$ equals 40. The input size is $1\times 40\times 200$.
>
> **Q5: In a real setting, how can we know if the prior dynamics are biased or not?**
>
> **A5:** In real settings, especially in the meteorological domains, the prior dynamics are almost always biased.  The point is that if the bias of the prior dynamics is within tolerance, we can consider them to be unbiased. This is exactly what has been done in the strong-constraint 4DVar formulation of data assimilation. However, if we want to achieve a more accurate assimilation result, the bias of prior dynamics should be considered [2]. This is why the weak-constraint 4DVar comes into being.
>
> **Q6: Is the white Gaussian noise assumption for observations reductive?**
>
> **A6:** No, such an assumption is not reductive. In the data assimilation problem, the error of observations can be divided into three parts: instrumental error, representative error, and conversion error [3]. The instrumental error represents the error introduced during measurement. The representative error refers to the error in interpolating the observatory data to the grid points. The conversion error refers to the error in converting the physical variable to observational variables. The only error we consider is the instrumental error. It is well known that the instrumental error can be formulated as Gaussian noise. We don't consider the representative error and conversion error because 1) the observations are on the grid points; 2) the observational variables and physical variables are identical. The simplification that omits representative error and conversion error is widely used in the data assimilation community, especially for developing novel data assimilation algorithms [4,5,6].
>
> **Q7: Why stop at 8 and 10 blocks for the experiments?**
>
> **A7:** It is OK to train the framework with more than 8 or 10 blocks. However, as the number of blocks gets larger, the training cost increases linearly, while the accuracy gain gradually saturates. We stopped at 8 and 10 blocks because we wanted to trade-off between the training cost and accuracy gain.

---

> ### Author Response · Authors · 2022-11-18
> **Response to Reviewer BfUY (3/3)**
>
> **Q8: Differences with Fablet et al. 2021**
>
> **A8:** The baseline method [1] defines an objective function that integrates both the observational information and the prior dynamics. It then adopts the idea of optimizer learning to build up recurrent neural networks to learn an optimizer for such an objective function. Each section of the recurrent neural network can be regarded as one step of optimization. Both the baseline and our proposed method progressively refine the outputs and maintain the dimensionality at each step. The major differences between our model and the baseline are shown below:
>
> 1. In [1], the blocks in neural networks are generally constructed based on the theory of gradient descent, whereas the three operators in our model are inspired by the intrinsic characteristics of the data assimilation problem.
> 2. Optimizer learning algorithms are applicable to all optimization problems theoretically, while our method is mainly applicable to the data assimilation problem. For the specific problem of data assimilation, experimental results demonstrate that our method can better combine the observational information with the prior dynamics, thus outperforming the optimizer learning-based baseline.
>
> **Q9: The color of Figure 3 does not match Figure 1**
>
> **A9:** In the revised version, we will replace Figure 3 with an algorithm box to illustrate our training procedure more clearly.
>
> **Q10: Reproducibility and codes**
>
> **A10:** Our framework is reproducible, and the training codes and our trained models will be released on GitHub.
>
> **Q11: Typos and writing issues**
>
> **A11:** Thanks for your suggestions. We will correct the typos and improve our clarity in the revised version.
>
> [1] Fablet, Ronan, et al. "Learning variational data assimilation models and solvers." *Journal of Advances in Modeling Earth Systems* 13.10 (2021): e2021MS002572.
>
> [2] Tr'emolet, Yannick. "Accounting for an imperfect model in 4D‐Var." *Quarterly Journal of the Royal Meteorological Society: A journal of the atmospheric sciences, applied meteorology and physical oceanography* 132.621 (2006): 2483-2504.
>
> [3] Kalnay, Eugenia. *Atmospheric modeling, data assimilation and predictability*. Cambridge university press, 2003.
>
> [4] Arcucci, Rossella, et al. "Deep data assimilation: integrating deep learning with data assimilation." *Applied Sciences* 11.3 (2021): 1114.
>
> [5] Bonavita, Massimo, and Patrick Laloyaux. "Machine learning for model error inference and correction." *Journal of Advances in Modeling Earth Systems* 12.12 (2020): e2020MS002232.
>
> [6] Brajard, Julien, et al. "Combining data assimilation and machine learning to emulate a dynamical model from sparse and noisy observations: A case study with the Lorenz 96 model." *Journal of Computational Science* 44 (2020): 101171.

---

> ### Comment · Reviewer_BfUY · 2022-12-12
> **thank you for the answers**
>
> I am keeping my grade (8).

---

### Official Review · Reviewer_wimX · 2022-10-23

**Confidence:** 3
**Correctness:** 3
**Technical Novelty And Significance:** 3
**Empirical Novelty And Significance:** 2
**Recommendation:** 6

**Clarity, Quality, Novelty And Reproducibility:**

Basically the writing is not bad, while some points are unclear as I commented above.

The quality of the paper looks to meet a minimum standard but could be improved much by a more intensive empirical study.

The specific learning procedure of the proposed method might be somewhat novel, if not overly impressive, but the current empirical materials provided in the paper are not sufficient to assess its utility. In particular, it lacks ablation studies to analyze the specific benefit of the proposed learning procedure.

I cannot really assess the reproducibility. Although the codes are not provided yet, it does not look difficult to implement all the things described in the paper from scratch.

**Strength And Weaknesses:**

### Strengths

- Method is simple.
- Improvement is clearly shown, compared to at least one baseline.

### Weaknesses

1. The motivation behind the design choice of the proposed method is not overly clear. In the caption of Figure 3, the authors explain that the proposed learning procedure is necessary for making the models perform as they expect. Though I agree with it per se, I do not think it explains the fundamental motivation. In particular, why do you want to iteratively apply the $\mathcal{F}$ and $\mathcal{P}$? How is the finetune-based procedure beneficial compared to learning all the networks at once?

2. The experiments lack comparison to an obvious baseline, that is, a network comprising UNet blocks trained altogether. This lack of comparison makes it difficult to assess the particular benefit of the proposed learning procedure.

3. Solving state estimation problems just like data augmentation using neural nets has been an active research area; e.g.:

- Krishnan+, Structured inference networks for nonlinear state space models, AAAI 2017
- Karl+, Deep variational Bayes filters: Unsupervised learning of state space models from raw data, ICLR 2017
- Marino+, A general method for amortizing variational filtering, NeurIPS 2018

among many others. However, the current paper does not address the potential relation to these kinds of studies. Maybe the authors' interest does not lie in Bayesian inference, but references from the outside of the data augmentation community would be highly valuable to make the paper more complete.

---

### Minor points

- Naming a network an "observation operator" without cautionary statements may be confusing especially for machine learning researchers. A natural interpretation is that such an operator takes the hidden state $x$ as input and gives the observation $y$ as output. But, what could be read from the paper is the opposite, that is, it takes $y$ as input and spits out an estimation of $x$.
- In the "Modified U-Net Structure" part, how you do the dimensionality reduction is unclear.

**Summary Of The Paper:**

A network architecture and the corresponding training strategy for solving data assimilation problems are proposed.

**Summary Of The Review:**

While the direction of the paper looks nice, the current empirical study does not tell much about the specific benefit of the proposed learning procedure, which makes the paper incomplete.

---

> ### Author Response · Authors · 2022-11-18
> **Response to Reviewer wimX (1/3)**
>
> We sincerely appreciate your careful and thoughtful comments and time. We try to explain your concerns point by point.
>
> **Q1: Motivation**
>
> **A1:** The crux of data assimilation is to combine the observational information and the prior dynamical information to reconstruct the physical states. A direct idea for solving this issue is to build a neural network, named the observation operator, to map observations to physical states. In order to integrate the prior dynamical information, a naive method is to add the dynamic loss to the loss function. However, such a method does not work well, which indicates that merely enforcing dynamic information in a posteriori manner is generally ineffective without the cooperation of the network framework.
>
> The motivation is that we can design a neural network whose function is to specifically ensure that the output conforms to the dynamical characteristics. We name the neural network "flow operator". In order to learn information from observations and labels (ground truth states), we further design the perturbator to work with the flow operator. The traditional data assimilation method involves both the spatial interpolation of observations and the prior dynamics. In our paper, the observation operator aims at the spatial interpolation of observations, while the perturbator and the flow operator mainly aim at integrating the prior dynamics. Such a framework can better integrate dynamic information than those done merely in a posteriori manner.

---

> ### Author Response · Authors · 2022-11-18
> **Response to Reviewer wimX (2/3)**
>
> **Q2: Comparison to an obvious baseline**
>
> **A2:** As suggested, we build up an obvious baseline in the following way. The neural network architecture is exactly the same as the proposed method, with one observation operator and one flow operator in the front and pairs of perturbators and flow operators following behind. The only difference is that we do not train the framework according to the proposed learning procedure. Instead, we follow the classical manner of training recurrent neural networks:
>
> (1) Set the number of blocks we are going to train.
>
> (2) Train the observation operator and the flow operator in the front.
>
> (3) Add one pair of the perturbator and the flow operator to the back, and then train the entire network.
>
> (4) Repeat (3) until the number of pairs meets the requirement in (1).
>
> A hybrid loss of reconstruction loss and dynamic loss is applied throughout training. The results are reported in the two tables below (the first one is for Lorenz-63 and the second one is for Lorenz-96).
>
> As for the Lorenz-63 system, when the block number is small, for example, two blocks, there is no significant difference between our work and the baseline. Nevertheless, as the number of blocks gets larger, the advantage of our proposed method starts to emerge. The higher the number of blocks, the better our method outperforms the baseline. When the block number is eight, the reconstruction error of our method is over 10% smaller than the baseline.
>
> The advantage is even more obvious in the Lorenz-96 system. For the baseline, the R-score reduces by only 16.7% from 2 blocks to 10 blocks; for our work, the R-score reduces by 35.7%. The R-score even rises when the number of blocks is increased from eight to ten for the baseline, which proves that it is easier for the classical method to overfit.  When the number of blocks is ten, the R-score of our method is 39.3% smaller than the baseline, and the dynamic loss is one order of magnitude smaller than the baseline.
>
> | Training procedure | # of blocks | R-score  | LossDyn  | Training procedure | # of blocks | R-score  | LossDyn  |
> | ------------------ | ----------- | -------- | -------- | ------------------ | ----------- | -------- | -------- |
> | Our work           | 2           | 1.28E+00 | 1.10E-03 | Baseline           | 2           | 1.18E+00 | 1.06E-03 |
> |                    | 4           | 9.23E-01 | 3.82E-04 |                    | 4           | 9.82E-01 | 6.03E-04 |
> |                    | 6           | 7.99E-01 | 2.08E-04 |                    | 6           | 8.65E-01 | 4.28E-04 |
> |                    | 8           | 7.51E-01 | 1.52E-04 |                    | 8           | 8.49E-01 | 3.99E-04 |
>
> | Training procedure | # of blocks | R-score  | LossDyn  | Training procedure | # of blocks | R-score  | LossDyn  |
> | ------------------ | ----------- | -------- | -------- | ------------------ | ----------- | -------- | -------- |
> | Our work           | 2           | 5.71E-01 | 1.40E-03 | Baseline           | 2           | 7.27E-01 | 4.32E-03 |
> |                    | 4           | 4.50E-01 | 5.09E-04 |                    | 4           | 6.54E-01 | 3.09E-03 |
> |                    | 6           | 3.97E-01 | 3.14E-04 |                    | 6           | 6.21E-01 | 2.80E-03 |
> |                    | 8           | 3.73E-01 | 2.66E-04 |                    | 8           | 5.99E-01 | 2.24E-03 |
> |                    | 10          | 3.67E-01 | 2.36E-04 |                    | 10          | 6.05E-01 | 2.37E-03 |
>
> In conclusion, the comparison to the baseline proves that our proposed training procedure has at least the following advantages:
>
> 1) Under the condition of the same number of blocks, lower reconstruction error and dynamic loss can be achieved.
> 2) More accuracy gain can be obtained when the number of blocks gets larger.
> 3) The proposed method is less prone to overfitting when the number of blocks becomes large.
>
> **Q3: Relationship with other state estimation problems**
>
> **A3:** Thanks for your suggestions. We have carefully read the three papers [2,3,4] you recommended, and we will refer to them in the new version. According to our understanding, the Bayes variation methods introduce the prior knowledge and make the inference based on the Bayes theory. Our method also introduces prior knowledge by training the perturbators and the flow operators using an unsupervised method. The Bayes variation method has strong robustness and interpretability. We will consider the advantage to further improve our model in future work.
>
> **Q4: Use of the term "observation operator"**
>
> **A4:** Thanks for pointing this out. We will use the term "inverse observation operator"[1] in the revised version instead.
>
> **Q5: The dimensionality reduction**
>
> **A5:** As shown in the legend of Figure 2 in the old version or Figure 4 in the revised version, the dimensionality reduction is done by max-pooling.

---

> ### Author Response · Authors · 2022-11-18
> **Response to Reviewer wimX (3/3)**
>
> **Q6: Ablation studies to analyze the specific benefit of the proposed learning procedure**
>
> **A6:** In addition to experiments comparing to an obvious baseline, we add two more ablation experiments to show that 1) the refining process with the perturbator and the flow operator greatly contributes to reducing the reconstruction error of the states generated by the observation operator; 2) when the experimental setting for observations is changed, only the observation operator needs retraining, and the network parameters for the perturbator and the flow operator can remain unchanged.
>
> *Experiment 1:* We remove all the perturbators and flow operators, and test the performance of one single observation operator. It has been found that the checkerboard-style refining process can further reduce the reconstruction errors for Lorenz systems by at least 50%, compared with one single observation operator. The detailed results are shown in the table below.
>
> | Dynamic system | Network structure             | R-score | LossDyn  |
> | -------------- | ----------------------------- | ------- | -------- |
> | Lorenz-63      | Observation operator only     | 1.95    | >1.00e-2 |
> |                | Our work with three operators | 0.751   | 1.52e-4  |
> | Lorenz-96      | Observation operator only     | 0.763   | >1.00e-2 |
> |                | Our work with three operators | 0.367   | 2.36e-4  |
>
> *Experiment 2:* Taking the Lorenz-96 system as an example, we design two different settings for observations. For the first setting, the observations are only a random selection of half of the components in the Lorenz system and we sample the observations every four steps; For the second setting, the observations are sampled every four steps for the components whose indexes are even. We conduct the experiments according to the following steps.
>
> Step 1: Train the framework with all three operators under the first experimental setting. We use it as a well-trained reference model.
>
> Step 2: Only train the observation operator under the second experimental setting. We use it to simulate the case where only the observation operator is trained after the observation changes
>
> Step 3: Concatenate the observation operator trained in Step 2 with the perturbator and the flow operator trained in Step 1 without further training.
>
> Step 4: Test the combined network under the second experimental setting.
>
> The results are reported in the following table. Meanwhile, we also report the results of another experiment, in which the observation settings are switched, and the same experiment procedure is conducted again with the above steps. It can be seen that when the perturbator and the flow operator are transferred to another observation setting, they still work well, despite a slight increase in the reconstruction error (smaller than 10%). It also shows that the function of the observation operator and that of the perturbator and the flow operator can be decomposed well.
>
> | Observation setting | In which observation setting the perturbator and flow operator are trained | R-score | LossDyn |
> | ------------------- | ------------------------------------------------------------ | ------- | ------- |
> | Random              | Random                                                       | 0.367   | 2.36e-4 |
> |                     | Even                                                         | 0.402   | 3.61e-4 |
> | Even                | Random                                                       | 0.304   | 3.61e-4 |
> |                     | Even                                                         | 0.291   | 2.56e-4 |
>
> **Q7: Reproducibility and codes**
>
> **A7:** Our framework is reproducible, and the training codes and our trained models will be released on GitHub.
>
> [1] Frerix, Thomas, et al. "Variational data assimilation with a learned inverse observation operator." *International Conference on Machine Learning*. PMLR, 2021.
>
> [2] Krishnan, Rahul, Uri Shalit, and David Sontag. "Structured inference networks for nonlinear state space models." *Proceedings of the AAAI Conference on Artificial Intelligence*. Vol. 31. No. 1. 2017.
>
> [3] Karl, Maximilian, et al. "Deep variational bayes filters: Unsupervised learning of state space models from raw data." *arXiv preprint arXiv:1605.06432* (2016).
>
> [4] Marino, Joseph, Milan Cvitkovic, and Yisong Yue. "A general method for amortizing variational filtering." *Advances in neural information processing systems* 31 (2018).

---

> > ### Comment · Reviewer_wimX · 2022-11-28
> > **Thanks for clarification**
> >
> > Thank you for the clarification. The new experimental results are nice to show the utility of the specific design choice of the method! I now increased the score.

---

### Official Review · Reviewer_6BrX · 2022-11-02

**Confidence:** 4
**Clarity, Quality, Novelty And Reproducibility:** See above
**Correctness:** 2
**Technical Novelty And Significance:** 3
**Empirical Novelty And Significance:** 2
**Recommendation:** 3

**Strength And Weaknesses:**

See above

**Summary Of The Paper:**

## Summary

The authors propose a new end-to-end fully data-driven method for data assimilation in which the different components of the system are fully parameterized using neural networks. Their model enjoys good accuracy and across different dynamical system parameterizations, specifically on the classical Lorenz systems.

## Review Summary

 Promising work but requires significant improvements.

1. Please clarify the role of prior dynamics in the model / training procedure.
2. The claims regarding state-of-the-art, especially regarding 4DVar should be weakened (1) there is no explicit comparison to 4DVar, (2) there are no experiments approaching the scale or real-world applications in which 4DVar is used. The first pages of the paper read as if the authors recommend this as a substitute.

## High-level Feedback

- "Complicated modeling and solving process of 4DVar..."
	- If this is an important model, please remind the read of the basic outlines of how this model works. Ideally, within the mathematical framework that you use to define your model.
- The definition of prior dynamics need to be defined early so that the paper makes sense. It is very clearly defined, but only in the results section of the paper. Essentially, it is the parameters of the dynamical system in question.
	- The experimental setup regarding generalization across prior dynamics needs to be clarified. Either this or the training setup. It's hard to know without answering the question: How do the prior dynamics enter your model / training setup? This key question should be clarified multiple times, e.g. in text as well as in diagrams and figures.
- Please make this explicit: Your argument is that you have superior accuracy and generalization across different systems (e.g. prior dynamics). I believe you are missing stability, an important property of numerical simulations. You don't have to tackle this, but please mention it. Ideally, include an analysis of the stability of your learned system over long unrolls, even if brief.

## Detailed Feedback

- Equation 2 is sloppy because it omits time. All numerical simulations of chaotic systems diverge for large enough $t$. So what exactly does $C$ measure?
- "Our main goal is to construct an inverse map, implemented in the form of a neural network [...]"
	- Yes, great. Please provide citations for this general idea. I was first exposed to it in the work of Max Welling and his collaborators who used this idea for Bayesian inference to map observations to latent variables. You may need to dig deeper to find the right sources.
		- https://arxiv.org/pdf/1906.02691.pdf
- Equation 6: What is the utility of this notation if ultimately $\Phi$ is simply a neural network?
	- How exactly do the prior dynamics come into play here?
- Equation 7: Are you saying here that you want $\mathcal F(x)$ to match the dynamics?
	- Why do you want to do this if you already have an implementation of the dynamics?
	- What if the function of the other terms in this equation?
	- Are you somehow trying to beat the numerical errors produced by the solver? Or this is where you get a mismatch between prior dynamics and your model (to be minimised during training).
- Equation 8: This is a neural ODE model. Maybe it is isn't exactly the same as the original paper, but it is in that spirit. Please make mention of this.
- Equation 9: Where is $\mathcal L_{dyn-pr}$ defined?
	- This entire paragraph is strange. Please compare and contrast with a simpler classical neural network training.
- Function of the Perturbator:
	- What does perturbation have to do with mapping states to observations? Or is it vice-versa? Is this a classical notion in physics? Is there any previous work / grounding of this idea? If the Perturbator is simply a network, how do you get multiple states out of single input (without adding noise somewhere in the process).
- Numerical Experiments
	- Time-to-solution: what is this? Training time? Unroll time?
	- Replace GENN aG-Conv with the citation, e.g. Fablet et al. 2021.
	- You seem to use a few tricks: stopping condition, pretraining, and perhaps more. Please clarify the improvements gained by each trick using ablation experiments. Knowing what did not work is very useful to the community and provides insight into the decisions that you made for this work.
	- If you did 10 independent tests, please provide std for all values.
	- Undefined terms: assimilation window, random half of the components, a variance of 2.0 is involved (what does this mean??)
	- Please move the basic properties of the dataset in the appendix to the main paper: number of examples, train / test / validation setup, etc.
	- Please move all figures after where they are first mentioned in the text. It is very frustrating to see a figure with undefined features only to discover that it is described in some random section below.
	- Figure 4:
		- Don't use "3 blocks," "4 blocks," etc. if what you are actually testing here is the role (e.g. smoothing / unsmoothing) of flow and perturbation operators.
		- The legend is laughably small.
		- This blue arrow seems to be guiding the reader about how to read the figure. Would be easier to just make a simpler figure which is more legible.
		- The MSE plot should be separate.
			- Label the y-axis "mean-squared error"
			- Remove the title which has too many words
			- Add legend
	- Figure 5
		- R-score means something else, not reconstruction loss.
			- https://en.wikipedia.org/wiki/R_score
		- You have already define $\mathcal L_{rec}$ , just use that.
- Appendix
	- Figure 6
		- Is this the dataset? I'm confused. A simple description, e.g. 70 / 20 / 10 % train / validation / test split would suffice.
			- Anyway, as mentioned above, this should go into the main paper.
	- Figure 7
		- What is happening when you switch to fine-tuning? Why this blow up?
	- Figure 9
		- From 6 to 10 blocks there is little change in error. What is happening here? Shouldn't you be overfitting as model complexity grows?
	- Does not belong in this part of my review but: What do you mean by self-supervised loss? Please clarify this since it probably does not match up with the classical ML usage of this term.

**Summary Of The Review:**

See above

---

> ### Author Response · Authors · 2022-11-18
> **Response to Reviewer 6BrX (1/4)**
>
> We sincerely appreciate your careful and thoughtful comments and time. We try to explain your concerns point by point, and the order of our answers is rearranged according to their importance.
>
> **Q1: Relationship with the 4DVar**
>
> **A1:** In this paper, we are dealing with the problem of data assimilation. The 4DVar algorithm is one of the most effective algorithms for data assimilation, especially in the area of numerical weather forecasting. Despite its success, the 4DVar algorithm still has the following disadvantages: 1) it has a relatively high computational cost; 2) it requires the development of the so-called "adjoint model", which brings a huge amount of engineering effort. Our work and related works [2, 3] aim to develop neural network-based algorithms for data assimilation to 1) reduce computational cost and 2) save the development effort. We admit that, at present, our work cannot exceed the 4DVar algorithm in terms of accuracy, because the training of our framework requires labels, which in real-world settings need to be provided by traditional algorithms. Instead, by comparison with [2], we demonstrate that our work is more efficient and accurate than another state-of-the-art network framework.
>
> We will try our best to clarify this in the revised version of our paper.
>
> **Q2: Why not define the problem within the 4DVar framework?**
>
> **Q2:** The goal of our work is to resolve the data assimilation problem. As a result, we define the problem of data assimilation in a more direct way and do not define the problem within the 4DVar framework since it is only a particular algorithm.
>
> **Q3: The role of prior dynamics in the model/training procedure.**
> **A3:** In the problem of data assimilation, all considered "dynamics" are the prior dynamics. Through data assimilation, we combine the prior dynamics and the observations to obtain the reconstructed states. We will clarify this in the revised version.
>
> In our framework, the prior dynamics are learned during the training process. To be specific, we use the dynamic loss to train the flow operator, and the calculation of prior dynamic loss requires the prior dynamics $\mathcal{M_{pr}}$. When we use it to train the flow operator, the prior dynamics are learned.
>
> **Q4: The stability of the learned system over long unrolls**
>
> **A4:** Thanks for your reminder. Data assimilation does not deal with the problem of learning the full dynamical process, and it is rather an optimization problem that fuses observations and prior dynamics. Our work is also constrained by stability, and the stable prior dynamic process is crucial for our work. The experiments we have performed so far have been based on stable prior dynamics and no collapse has occurred. In the future, when our framework is extended to more complex dynamics, we need to consider the stability issue.
>
> **Q5: "Equation 6: What is the utility of this notation if ultimately Φ is simply a neural network?"**
>
> **A5:** $\Phi$ is not implemented by a neural network. The definition of $\Phi$ is somewhat imprecise in our original paper, which led to some misunderstandings. The precise definition of $\Phi$ should be
> $$
> \Phi(x)(t)=x(t-\Delta t) +\int_{t-\Delta t}^{t}\mathcal{M_{pr}}(x(u))du
> $$
> It is essentially an intermediate operator for calculating the dynamic loss. As long as the prior dynamic integration (by numerical solver) is given, $\Phi$ is determined. No neural network is involved here.
>
> **Q6: Where is $\mathcal{L_{dyn-pr}}$ defined?**
>
> **A6:** Sorry for the unclear definition and the confusion it creates.  $\mathcal{L_{dyn-pr}}$ shares exactly the same meaning with $\mathcal{L_{dyn}}$. We add the subscript "pr" here to emphasize that we use the prior dynamics to calculate the dynamic loss. In fact, we use the prior dynamics throughout the paper. We will clarify this in the revised version.
>
> **Q7: Meaning of Equation 2**
>
> **A7:** We only consider the error of one-step integration here. $\mathcal{M}$ represents the gradient in the current step for integration. We require that the difference between gradients be bounded to ensure that the prior dynamics are qualified to estimate the ground truth dynamics.

---

> ### Author Response · Authors · 2022-11-18
> **Response to Reviewer 6BrX (2/4)**
>
> **Q8: "Equation 7: Are you saying here that you want F(x) to match the dynamics?"**
>
> **1. Why do you want to do this if you already have an implementation of the dynamics?**
>
> First, we cannot obtain an accurate initial state to integrate with. Our work deals with the difficulty in a different method: we first use the observation operator to simultaneously generate a time sequence of coarse states and then use the flow operator and the perturbator for refinement. The role of the flow operator is to input a complete time series and refine it to make it more consistent with the prior dynamics.
>
> Second, the prior dynamics can be biased, and our framework aims to assimilate under biased prior dynamics.
>
> **2. What is the function of the other terms in this equation?**
>
> When the dynamic loss becomes smaller than $\mathcal{L_0}$ , we stop training and turn to the next phase. In practice, we don't actually set an $\mathcal{L_0}$, we fix the number of epochs instead. $\mathcal{L_0}$ is used here as an upper bound to show that after the operation of the flow operator, the dynamic loss is greatly reduced.
>
> **3. Are you somehow trying to beat the numerical errors produced by the solver? Or this is where you get a mismatch between prior dynamics and your model (to be minimized during training).**
>
> The first statement is not true; we are not trying to beat the numerical errors produced by the solver. The second statement is right; we want to minimize the mismatch between the prior dynamics and our model during training while keeping the observation characteristics as much as possible.
>
> **Q9: This entire paragraph (Network Training and Stopping Criterion) is strange.**
>
> **A9:** As stated above, we want to use the flow operator to minimize the mismatch between the prior dynamics and our model during training. However, if the prior dynamics are biased, we cannot learn too much from the prior dynamics. This is why we set the stopping criterion: to avoid overfitting the prior dynamics.
>
> **Q10: Function of the Perturbator**
>
> The function of the perturbator, in short, is to help the flow operator better integrate the prior dynamics into the reconstructed states.
>
> **1. What does perturbation have to do with mapping states to observations? Or is it vice-versa?**
>
> Both the inputs and outputs of the perturbator are states.
>
> **2. Is this a classical notion in physics? Is there any previous work/grounding of this idea?**
>
> We think that this idea stems from the intrinsic characteristics of data assimilation. The traditional data assimilation method involves both the spatial interpolation of observations and the prior dynamics. In this paper, the observation operator aims at the spatial interpolation of observations, while the perturbator and the flow operator mainly aim at integrating the prior dynamics. Such a framework can better integrate dynamic information than others.
>
> **3. If the Perturbator is simply a network, how do you get multiple states out of single input?**
>
> We don't need to get multiple outputs. The perturbator produces only one deterministic perturbation.
>
> **Q11: Citation for the idea of the inverse map**
>
> **A11:** Thanks. We will add the citation in the revised version.
>
> **Q12: "Equation 8: This is a neural ODE model. Maybe it isn't exactly the same as the original paper, but it is in that spirit. Please make mention of this."**
>
> **A12:** We don't think it is a neural ODE model because the output of a neural ODE is one step ahead of the input, whereas both the input and the output of our flow operator are a sequence of physical states, corresponding to the same time window.
>
> **Q13: Time-to-solution: what is this? Training time? Unroll time?**
>
> **A13:** Time-to-solution refers to the time for the neural network to finish evaluation for one round on the test set. Specifically, in our experiments, we fix the batch size to eight for all the test experiments to ensure fairness. The shorter the test time, the faster our model runs.
>
> **Q14: Replace GENN aG-Conv with the citation, e.g. Fablet et al. 2021.**
>
> **A14:** We will use the term "Fablet et al. (2021)" to refer to the work of [1] in the revised version of our paper.

---

> ### Author Response · Authors · 2022-11-18
> **Response to Reviewer 6BrX (3/4)**
>
> **Q15: If you did 10 independent tests, please provide std for all values.**
>
> **A15:** We repeat ten experiments on the same dataset. The reconstruction loss and the dynamic loss are unchanged in the experiments because there is no randomness in our framework once it is trained. The computing time of ten independent tests shows some variance because we conduct our experiments on the publicly used computing cluster of our lab. We report the test-time results in the tables below (see the above one for Lorenz-63 and the below one for Lorenz-96, the unit of the results is second). We will add the results of the standard deviations in the revised version.
>
> |         | Our work (2 blocks) | Our work (4 blocks) | Our work (6 blocks) | Our work (8 blocks) | Fablet et al. (2021) |
> | ------- | ------------------- | ------------------- | ------------------- | ------------------- | -------------------- |
> | Average | 3.06                | 4.51                | 5.82                | 6.40                | 27.10                |
> | Std dev | 0.65                | 0.48                | 0.63                | 0.40                | 3.80                 |
> | Exp 1   | 3.50                | 4.03                | 5.31                | 6.31                | 30.64                |
> | Exp 2   | 3.74                | 4.78                | 6.80                | 6.91                | 32.66                |
> | Exp 3   | 3.67                | 4.62                | 5.39                | 6.11                | 28.07                |
> | Exp 4   | 2.45                | 4.45                | 5.24                | 6.01                | 28.20                |
> | Exp 5   | 2.42                | 4.33                | 5.17                | 6.99                | 22.84                |
> | Exp 6   | 2.47                | 4.07                | 5.85                | 6.96                | 24.76                |
> | Exp 7   | 3.83                | 4.68                | 6.51                | 6.26                | 23.72                |
> | Exp 8   | 2.45                | 4.31                | 6.02                | 6.06                | 21.84                |
> | Exp 9   | 2.47                | 5.66                | 6.60                | 6.25                | 26.50                |
> | Exp 10  | 3.57                | 4.20                | 5.29                | 6.12                | 31.72                |
>
> |         | Our work (2 blocks) | Our work (4 blocks) | Our work (6 blocks) | Our work (8 blocks) | Our work (10 blocks) | Fablet et al. (2021) |
> | ------- | ------------------- | ------------------- | ------------------- | ------------------- | -------------------- | -------------------- |
> | Average | 7.76                | 9.77                | 11.45               | 13.46               | 15.86                | 47.29                |
> | Std dev | 0.60                | 0.97                | 0.93                | 0.77                | 0.72                 | 1.46                 |
> | Exp 1   | 7.88                | 10.06               | 11.94               | 14.71               | 15.13                | 47.99                |
> | Exp 2   | 7.89                | 9.18                | 10.23               | 12.45               | 16.93                | 48.00                |
> | Exp 3   | 8.58                | 8.41                | 11.04               | 12.45               | 16.55                | 47.70                |
> | Exp 4   | 7.24                | 10.59               | 11.37               | 13.28               | 15.36                | 46.73                |
> | Exp 5   | 7.23                | 9.80                | 12.54               | 13.05               | 15.84                | 44.41                |
> | Exp 6   | 8.96                | 10.52               | 11.94               | 14.63               | 15.35                | 47.71                |
> | Exp 7   | 7.21                | 10.92               | 12.97               | 13.51               | 16.88                | 45.02                |
> | Exp 8   | 7.31                | 9.27                | 10.45               | 13.28               | 15.28                | 48.17                |
> | Exp 9   | 7.78                | 10.76               | 11.62               | 13.62               | 15.16                | 48.64                |
> | Exp 10  | 7.50                | 8.22                | 10.42               | 13.65               | 16.11                | 48.51                |

---

> ### Author Response · Authors · 2022-11-18
> **Response to Reviewer 6BrX (4/4)**
>
> **Q16: The function of the tricks, such as stopping condition and pretraining.**
>
> **A16:** The stopping condition is crucial to avoid overfitting the prior dynamics. By removing the stopping condition, our framework would be theoretically flawed, and the training loss for the perturbator would diverge fast.
>
> In the old version of our paper, the term "pretraining" has a different meaning from, for example, neural language processing. We use the term "pretraining" to refer to the training phase in which the parameters of the observation operator and the flow operator are trained. During both "pretraining" and "finetuning", our model is trained with the same dataset.  Therefore, pretraining is not a trick but just a phase in our checker-board style training procedure. By removing the "pretraining" procedure, our framework will not converge because the training procedure is incomplete. In our revised version, we will clarify our training procedure more clearly.
>
> **Q17: Clarify undefined terms: assimilation window, a random half of the components, a variance of 2.0 is involved**
>
> **A17:** Thanks for pointing out these. The assimilation window refers to the time period of interest for assimilation. A random half of the components means that the observations are only a random selection of half of the components in the Lorenz systems. A variance of 2.0 is involved means that the variance of the Gaussian noise for observation is 2.0. They will be clarified in the revised version.
>
> **Q18: R-score means something else, not reconstruction loss.**
>
> **A18:** We will avoid using this term in the revised version.
>
> **Q19: Is Figure 6 the dataset?**
>
> **A19:** Yes, this is the dataset for training and evaluation [4].
>
> **Q20: What is happening when you switch to fine-tuning? Why does this blow up?**
>
> **A20:** As is shown in the figure of the training procedure, in the first step of fine-tuning, we add the perturbator to the framework, which has never been trained before. This leads to a sudden increase in loss. We will clarify this in the revised version.
>
> **Q21: From 6 to 10 blocks there is little change in error. What is happening here? Shouldn't you be overfitting as model complexity grows?**
>
> **A21:** Although the difference between 6 blocks and 10 blocks is small, the reconstruction loss drops nearly 10%, from 0.397 to 0.367. The accuracy gain is smaller than that from 2 blocks to 6 blocks because as the number of blocks gets larger, less useful information can be additionally provided and added by the perturbator and the flow operator. When the number of blocks is too large, it will lead to overfitting. Hence, we cannot train our framework with too many blocks.
>
> **Q22: What do you mean by self-supervised loss?**
>
> **A22:** In the revised version of our paper, we will use the term "unsupervised" instead of "self-supervised".
>
> When we train the flow operator, we adopt the dynamic loss. The calculation of the dynamic loss requires no labels and is therefore unsupervised.
>
> **Q23: Writing issues**
>
> **A23:** Thanks for your suggestions. We will improve our clarity in the revised version. Specifically, we will 1) move the basic properties of the dataset in the appendix to the main paper, 2) move all figures after where they are first mentioned in the text, 3) redraw Figure 4 and Figure 5 to improve legibility, and 4) do some other modifications.
>
> [1] Tr'emolet, Yannick. "Accounting for an imperfect model in 4D‐Var." *Quarterly Journal of the Royal Meteorological Society: A journal of the atmospheric sciences, applied meteorology and physical oceanography* 132.621 (2006): 2483-2504.
>
> [2] Fablet, Ronan, et al. "Learning variational data assimilation models and solvers." *Journal of Advances in Modeling Earth Systems* 13.10 (2021): e2021MS002572.
>
> [3] Frerix, Thomas, et al. "Variational data assimilation with a learned inverse observation operator." *International Conference on Machine Learning*. PMLR, 2021.
>
> [4] Schultz, M. G., et al. "Can deep learning beat numerical weather prediction?." *Philosophical Transactions of the Royal Society A* 379.2194 (2021): 20200097.

---

### Official Review · Reviewer_oVQ6 · 2022-11-03

**Confidence:** 3
**Correctness:** 3
**Technical Novelty And Significance:** 3
**Empirical Novelty And Significance:** 2
**Recommendation:** 6

**Clarity, Quality, Novelty And Reproducibility:**

To my knowledge, the proposed method is novel. The novelty resides in the fact that the method is composed of three types of networks, allowing to flexibly balance the different problem constraints.

However, the quality of the paper could be improved (see section on weaknesses). In addition, the lack of clarity impedes the full appreciation of the quality and significance of the work. In particular, the paper contains a substantial amount of typos and imprecisions that I would appreciate the authors to address:

-"and much more novel and bolder". This wording is slightly subjective, so I would refrain from writing this;
-several instances where either the article "a" is missing or the plural form should be used: "The aim of learning observation operator is to use neural network", "and another group of works uses neural network", "first work that directly uses neural network to";
-"exists a constant C" rather than "exits a constant C";
-Figure 2 legend. After "denote", "for" should be dropped, so "We denote K the spatial..., T the total...T_o the sequence..., T_h the intermediate...";
-"To tackle this difference", instead of "To tackle with this difference";
-page 5, "The major function of the flow operator is to restore the reconstructed sequence of states with a relatively high dynamic loss". Shouldn't it be "low dynamic loss"?
-"We train the flow operator with the dynamic loss, and the training process is self-supervised because no additional label is required". The authors should be explicit about why the training is self-supervised, and remind the reader what is the data the network is being trained on;
-the authors should mathematically define what L_{dyn-pr(\hat x)} is;
-"As long as inequality 9 is not satisfied, we end the current training phase and move on to the next one". Up until this point in the manuscript, it is unclear that the method is composed of several training phases, so I would suggest the authors to clarify this early in the manuscript. Perhaps the authors could also add an algorithm box early in the manuscript, so that the reader can follow how each step fits into the whole algorithm;
-typo in "Specifically, The perturbator";
-in Table 5, the abbreviation "GENN aG-LSTM" should be introduced;
-"Lorenz-63 and Lorenz-96 ... Both systems are one-dimensional." The systems are not one-dimensional neither in terms of state variables nor in terms of parameters. What do the authors mean by "one-dimensional"?
-typo in "ground truth dynamic";
-in Figure 4, bottom left panel, please describe what the dashed blue line corresponds to and discuss it in the text;
-Figure 5,"The left panels show" and "The right panels show".

**Strength And Weaknesses:**

The paper is technically sound, previous work is cited and discussed, and claims are for the most part well supported by empirical evaluation. However, I have a few concerns I would appreciate the authors to address:

-Table 1 shows the average results over 10 independent tests. It would be important to also report the respective standard deviations and assess the empirical performance in light of these standard deviations (e.g., performing statistical tests);

-Table 1 reports test times. What is considered test times? On one hand, this should be clarified in the manuscript, and on the other hand, training times should also be reported (if reporting training times does not make sense, then this should be justified);

-Figure 5 quantifies the performance of the new method under different degrees of bias in the dynamics. While this is a useful experiment, it would be equally important to show how the state-of-the-art method performs under these same circumstances.

**Summary Of The Paper:**

This study proposes a new method for data assimilation, i.e., for learning model-constrained dynamics from empirical data and estimating the state of the system of study. The method is composed of three types of networks (the observation operator, the flow operator and the perturbator), allowing it to flexibly balance the problem of reconstruction of the system latent states and the conforming to the specified model dynamics. On two classical problems in weather sciences, the Lorenz-63 and Lorenz-96, the method is shown to achieve higher performance than the state-of-the-art method 4D-Var (Fablet et al. 2021) in terms of reconstruction, dynamics loss, and computational cost.

**Summary Of The Review:**

Overall, the paper is technically sound and has a novel methodological contribution with somewhat convincing empirical results. However, some of the empirical results need to be further expanded and the clarity of the manuscript could be improved.

---

> ### Author Response · Authors · 2022-11-18
> **Response to Reviewer oVQ6 (1/5)**
>
> We sincerely appreciate your careful and thoughtful comments and time. We try to explain your concerns point by point.
>
> **Q1:  Standard deviations of different tests and performance assessment**
>
> **A1:** We repeat ten experiments on the same dataset. The reconstruction loss and the dynamic loss are unchanged in the experiments because there is no randomness in our framework once it is trained. The computing time of ten independent tests shows some variance because we conduct our experiments on the publicly used computing cluster of our lab. We report the test-time results in the tables below (see the above one for Lorenz-63 and the below one for Lorenz-96, the unit of the results is second). We will add the results of the standard deviations in the revised version.
>
> |         | Our work (2 blocks) | Our work (4 blocks) | Our work (6 blocks) | Our work (8 blocks) | Fablet et al. (2021) |
> | ------- | ------------------- | ------------------- | ------------------- | ------------------- | -------------------- |
> | Average | 3.06                | 4.51                | 5.82                | 6.40                | 27.10                |
> | Std dev | 0.65                | 0.48                | 0.63                | 0.40                | 3.80                 |
> | Exp 1   | 3.50                | 4.03                | 5.31                | 6.31                | 30.64                |
> | Exp 2   | 3.74                | 4.78                | 6.80                | 6.91                | 32.66                |
> | Exp 3   | 3.67                | 4.62                | 5.39                | 6.11                | 28.07                |
> | Exp 4   | 2.45                | 4.45                | 5.24                | 6.01                | 28.20                |
> | Exp 5   | 2.42                | 4.33                | 5.17                | 6.99                | 22.84                |
> | Exp 6   | 2.47                | 4.07                | 5.85                | 6.96                | 24.76                |
> | Exp 7   | 3.83                | 4.68                | 6.51                | 6.26                | 23.72                |
> | Exp 8   | 2.45                | 4.31                | 6.02                | 6.06                | 21.84                |
> | Exp 9   | 2.47                | 5.66                | 6.60                | 6.25                | 26.50                |
> | Exp 10  | 3.57                | 4.20                | 5.29                | 6.12                | 31.72                |
>
> |         | Our work (2 blocks) | Our work (4 blocks) | Our work (6 blocks) | Our work (8 blocks) | Our work (10 blocks) | Fablet et al. (2021) |
> | ------- | ------------------- | ------------------- | ------------------- | ------------------- | -------------------- | -------------------- |
> | Average | 7.76                | 9.77                | 11.45               | 13.46               | 15.86                | 47.29                |
> | Std dev | 0.60                | 0.97                | 0.93                | 0.77                | 0.72                 | 1.46                 |
> | Exp 1   | 7.88                | 10.06               | 11.94               | 14.71               | 15.13                | 47.99                |
> | Exp 2   | 7.89                | 9.18                | 10.23               | 12.45               | 16.93                | 48.00                |
> | Exp 3   | 8.58                | 8.41                | 11.04               | 12.45               | 16.55                | 47.70                |
> | Exp 4   | 7.24                | 10.59               | 11.37               | 13.28               | 15.36                | 46.73                |
> | Exp 5   | 7.23                | 9.80                | 12.54               | 13.05               | 15.84                | 44.41                |
> | Exp 6   | 8.96                | 10.52               | 11.94               | 14.63               | 15.35                | 47.71                |
> | Exp 7   | 7.21                | 10.92               | 12.97               | 13.51               | 16.88                | 45.02                |
> | Exp 8   | 7.31                | 9.27                | 10.45               | 13.28               | 15.28                | 48.17                |
> | Exp 9   | 7.78                | 10.76               | 11.62               | 13.62               | 15.16                | 48.64                |
> | Exp 10  | 7.50                | 8.22                | 10.42               | 13.65               | 16.11                | 48.51                |

---

> ### Author Response · Authors · 2022-11-18
> **Response to Reviewer oVQ6 (2/5)**
>
> **A1 (cont.):** Moreover, in order to further analyze the efficiency improvement of our framework, we do two t-tests on 1) the test time of our work (8 blocks) and Fablet et al. (2021) for the Lorenz-63 system, 2) the test time of our work (10 blocks) and Fablet et al. (2021) for the Lorenz-96 system. Denote $\tau_{63}$ the test time of Fablet et al. (2021) minus that of our work (8 blocks) on the Lorenz-63 system and $\tau_{96}$ the test time of Fablet et al. (2021) minus that of our work (10 blocks) on the Lorenz-96 system. The results are shown below.
>
> The mean of $\tau_{63}$ equals 20.7 (seconds) and the 95% confidence interval of $\tau_{63}$ is $[18.2, 23.2]$ (seconds). The mean of $\tau_{96}$ equals 31.4 (seconds) and the 95% confidence interval is $[30.4, 32.5]$ (seconds). The results suggest that the test speed of our work is much faster than that of [1].
>
> **Q2:  Definition of the test time**
>
> **A2:** Conforming to the paradigm of machine learning research, we split our dataset into the training set, the evaluation set, and the test set. The test time refers to the time for the neural network to finish evaluation for one round on the test set. Specifically, in our experiments, we fix the batch size to eight for all the test experiments to ensure fairness. The shorter the test time, the faster our model runs.

---

> ### Author Response · Authors · 2022-11-18
> **Response to Reviewer oVQ6 (3/5)**
>
> **Q3: Training time report**
>
> **A3:** The training cost of our algorithm is affordable on one TITAN RTX GPU, and is also much smaller than that in [1].
>
> We report the detailed training cost in the two tables below (the first one is for Lorenz-63 and the second one is for Lorenz-96). If the validation loss does not continue to decrease, we consider the network to converge and end the current phase. The training procedure of Fablet et al. (2021) is divided into two phases: the first phase is trained with 10 blocks and the second phase is trained with 20 blocks. We should at least allocate 20 epochs for the first phase and 100 epochs for the second phase to make the training loss converge well. As for our work, the numbers of epochs for the phases in the tables correspond to those used in our paper.
>
> In the experiment of the Lorenz-63 system, our network takes approximately 70 minutes to converge, and [1] takes over 200 minutes. As for the Lorenz-96 system, the training time of our network is about 187 minutes, and the training time of [1] is over 520 minutes. We will include the results in our revised version.
>
> Training time for the Lorenz-63 system:
>
> |                          | Phase                          | Training time per epoch | # of epochs | Phase time |
> | ------------------------ | ------------------------------ | ----------------------- | ----------- | ---------- |
> | Fablet et al. (2021) [1] | Phase 1 (10 blocks)            | 1 m 38 s                | 20          | 32 m 40 s  |
> |                          | Phase 2 (20 blocks)            | 1 m 45 s                | 100         | 175 m      |
> |                          | Total time                     |                         |             | 207 m 40 s |
> | Our work                 | Train the observation operator | 7 s                     | 20          | 2 m 20 s   |
> |                          | Train the flow operator        | 10 s                    | 200         | 33 m 20 s  |
> |                          | Finetune 1 (4 blocks)          | 24 s                    | 20          | 8 m        |
> |                          | Finetune 2 (6 blocks)          | 36 s                    | 20          | 12 m       |
> |                          | Finetune 3 (8 blocks)          | 44 s                    | 20          | 14 m 40 s  |
> |                          | Total time                     |                         |             | 70 m 20 s  |
>
> Training time for the Lorenz-96 system:
>
> |                      | Phase                          | Training time per epoch | # of epochs | Time       |
> | -------------------- | ------------------------------ | ----------------------- | ----------- | ---------- |
> | Fablet et al. (2021) | Phase 1 (10 blocks)            | 2 m 34 s                | 20          | 51 m 20 s  |
> |                      | Phase 2 (20 blocks)            | 4 m 45 s                | 100         | 475 m      |
> |                      | Total time                     |                         |             | 526 m 20 s |
> | Our work             | Train the observation operator | 9 s                     | 20          | 3 m        |
> |                      | Train the flow operator        | 33 s                    | 200         | 110 m      |
> |                      | Finetune 1 (4 blocks)          | 33 s                    | 20          | 11 m       |
> |                      | Finetune 2 (6 blocks)          | 48 s                    | 20          | 16 m       |
> |                      | Finetune 3 (8 blocks)          | 63 s                    | 20          | 21 m       |
> |                      | Finetune 4 (10 blocks)         | 78 s                    | 20          | 26 m       |
> |                      | Total time                     |                         |             | 187 m      |

---

> > ### Comment · Reviewer_oVQ6 · 2022-12-12
> > **updated score**
> >
> > I would like to thank the authors for addressing the reviews, including mine, to some satisfaction. Given this, I will raise my score to 6.

---

> ### Author Response · Authors · 2022-11-18
> **Response to Reviewer oVQ6 (4/5)**
>
> **Q4: How does the state-of-the-art perform under biases?**
>
> **A4:** We add the experiments to compare the performance of our work and [1] under different degrees of biases. The prior dynamics construction, including the parameter selection of biases, follows the same way as in our paper. In the Lorenz-63 system, the biased prior dynamics are constructed by assigning $\Delta \sigma \in \{0, 0.5, 1, 1.5, 2, 2.5, 3, 3.5, 4, 4.5, 5\}$; in the Lorenz-96 system, the biased prior dynamics are constructed by assigning $\Delta F \in \{0, 0.5, 1, 1.5, 2, 2.5, 3, 3.5, 4, 4.5, 5\}$.
>
> The results are shown in the table below. It is worth noting that [1] did not realize their algorithms under biased prior dynamics, and we implement their algorithms based on our best understanding of their work. The results of the case in which bias equals zero are close to those presented in [1], which confirms the credibility of our experiments.
>
> As for the Lorenz-63 system, when the bias is larger than 1, the reconstruction loss of [1] ranges from 1.19 to 1.36, which is similar to the performance of our model. As for the Lorenz-96 system, the reconstruction loss of [1] seems to diverge as the prior bias gets larger, while the reconstruction loss of our model converges to around 0.7. These results show that our proposed model is more adaptive to the prior bias than [1].
>
> | bias (Delta sigma) | lossDyn (Our work) | R-score (Our work) | lossDyn (Fablet et al., 2021) | R-score (Fablet et al., 2021) |
> | ------------------ | ------------------ | ------------------ | ----------------------------- | ----------------------------- |
> | 0                  | 1.52E-04           | 7.51E-01           | 1.84E-02                      | 1.07E+00                      |
> | 0.5                | 2.60E-04           | 8.20E-01           | 1.32E-03                      | 1.10E+00                      |
> | 1                  | 6.15E-04           | 1.01E+00           | 1.20E-02                      | 1.28E+00                      |
> | 1.5                | 9.13E-04           | 1.09E+00           | 2.88E-02                      | 1.36E+00                      |
> | 2                  | 1.91E-03           | 1.17E+00           | 2.04E-02                      | 1.29E+00                      |
> | 2.5                | 2.14E-03           | 1.29E+00           | 1.74E-02                      | 1.33E+00                      |
> | 3                  | 3.33E-03           | 1.33E+00           | 1.18E-02                      | 1.27E+00                      |
> | 3.5                | 3.70E-03           | 1.26E+00           | 1.32E-02                      | 1.29E+00                      |
> | 4                  | 3.35E-03           | 1.19E+00           | 2.26E-02                      | 1.19E+00                      |
> | 4.5                | 5.35E-03           | 1.18E+00           | 1.58E-02                      | 1.20E+00                      |
> | 5                  | 7.44E-03           | 1.39E+00           | 1.65E-02                      | 1.23E+00                      |
>
> | bias (Delta F) | lossDyn (Our work) | R-score (Our work) | lossDyn (Fablet et al., 2021) | R-score (Fablet et al., 2021) |
> | -------------- | ------------------ | ------------------ | ----------------------------- | ----------------------------- |
> | 0              | 2.66E-04           | 3.73E-01           | 2.32E-02                      | 3.80E-01                      |
> | 0.5            | 5.29E-04           | 3.88E-01           | 1.58E-02                      | 4.82E-01                      |
> | 1              | 1.98E-03           | 5.16E-01           | 5.43E-02                      | 6.48E-01                      |
> | 1.5            | 4.32E-03           | 5.96E-01           | 5.18E-02                      | 7.74E-01                      |
> | 2              | 8.34E-03           | 6.68E-01           | 4.45E-02                      | 8.26E-01                      |
> | 2.5            | 1.36E-02           | 7.21E-01           | 4.04E-02                      | 9.41E-01                      |
> | 3              | 1.91E-02           | 6.86E-01           | 1.12E-02                      | 8.50E-01                      |
> | 3.5            | 2.59E-02           | 7.50E-01           | 1.34E-02                      | 9.55E-01                      |
> | 4              | 2.93E-02           | 7.40E-01           | 2.93E-02                      | 1.09E+00                      |
> | 4.5            | 3.94E-02           | 7.40E-01           | 2.10E-02                      | 1.20E+00                      |
> | 5              | 4.51E-02           | 7.70E-01           | 2.63E-02                      | 1.43E+00                      |

---

> ### Author Response · Authors · 2022-11-18
> **Response to Reviewer oVQ6 (5/5)**
>
> **Q5:  Writing issue**
>
> **A5:** Thank you for kindly pointing out the typos and imprecisions in our paper. We will improve the presentation accordingly in the revised version. We also want to address your concerns regarding the content of the paper here.
>
> **1. The wording "much more novel and bolder" is slightly subjective.**
>
> We will remove this subjective expression in the revised version.
>
> **2. "The major function of the flow operator is to restore the reconstructed sequence of states with a relatively high dynamic loss". Shouldn't it be "low dynamic loss"?**
>
> The dynamic loss of the outputs of the flow operator is lower than that of the inputs. To avoid ambiguity, we will rephrase the sentence into the following.
>
> "The main function of the flow operator is to reduce the dynamic loss of the reconstructed sequence of states and output a sequence that better conforms to the prior dynamics. "
>
> **3. Why the flow operator training is self-supervised?**
>
> In the revised version of our paper, we will use the term "unsupervised" instead of "self-supervised".
>
> When we train the flow operator, we adopt the dynamic loss. The calculation of the dynamic loss requires no labels and is therefore unsupervised.
>
> **4. Definition of $\mathcal{L}_{dyn-pr}$**
>
> $\mathcal{L_{dyn-pr}}$ shares exactly the same meaning with $\mathcal{L_{dyn}}$. We add the subscript "pr" here to emphasize that we use the prior dynamics rather than the ground truth dynamics to calculate the dynamic loss. In fact, we use the prior dynamics throughout the paper. We will clarify this in the revised version.
>
> **5. Clarify the training procedure in the manuscript**
>
> In the revised version, we will rearrange the order of the sections to explain our framework more clearly, and an algorithm box will also be added.
>
> **6. In Table 5, the abbreviation "GENN aG-LSTM" should be introduced.**
>
> We will use the term "Fablet et al. (2021)" to refer to the work of [1] in the revised version of the paper.
>
> **7. Why say that the systems are one-dimensional?**
>
> Thanks for pointing out this. By saying that the systems are one-dimensional, we mean that the physical dimension of the system is one. Take temperature modeling as an example. If we want to numerically model and solve for temperature on a 1-D line, we need to discretize the line into grid points. If there are totally $N$ grid points, the dimension of the discrete equation is $N$, but the corresponding physical problem has only one dimension. The Lorenz-96 system is generally considered to be physically one-dimensional, and the variables of the Lorenz-96 system can be regarded as discrete points on a circle. The relevant expressions will be clarified with discrete dimensions in the revised version.
>
> **8. "In Figure 4, bottom left panel, please describe what the dashed blue line corresponds to and discuss it in the text."**
>
> The dashed blue line corresponds to the change in reconstruction loss when blocks are added one by one. When the number of blocks is odd, the network ends up with the perturbator, resulting in a relatively high reconstruction loss. The bottom left panel of Figure 4 will be removed in the revised version due to low importance.
>
> [1] Fablet, Ronan, et al. "Learning variational data assimilation models and solvers." *Journal of Advances in Modeling Earth Systems* 13.10 (2021): e2021MS002572.

---

### Official Review · Reviewer_1MTc · 2022-11-04

**Confidence:** 4
**Correctness:** 4
**Technical Novelty And Significance:** 3
**Empirical Novelty And Significance:** 3
**Recommendation:** 8

**Clarity, Quality, Novelty And Reproducibility:**

The work is presented with a lot of clarity and focus on its scientific story. As such the writing is of consistently high quality with minor typos at times (see below), and has a significant amount of originality in the design of the operator learning approach. While the individual components, and their combination together are novel, the depth of the paper could be enhanced by relating the flow operator learning approach back to the iterative refinement used in e.g. diffusion models. The checker board-style training approach could also be further rooted in preceding literature. While not immediately comparable, highly similar approaches have appeared before and further references could help to underline the intellectual thread this training approach builds upon.

While the choice of using the U-Net architecture is widespread, and has shown good performance in literature, I would encourage the authors to include discussion of the work of Wang et al. [1], which showed that U-Nets are not always the best choice of network for chaotic dynamic / fluid dynamics, and might present a limitation of the presented approach when being extended to other more difficult chaotic systems.

[1] Wang, Rui, Karthik Kashinath, Mustafa Mustafa, Adrian Albert, and Rose Yu. "Towards physics-informed deep learning for turbulent flow prediction." In Proceedings of the 26th ACM SIGKDD International Conference on Knowledge Discovery & Data Mining, pp. 1457-1466. 2020.

Minor typos in the text:
- pg 1: Four-dimensional variational assimilation (..) -> The four-dimensional (..)
- pg 2: The aim of learning observation operator is (..) -> The aim of learning observation operators (..)
- pg 2: to use neural network to construct a map (..) -> to use neural networks to (..)
- pg 3: the bias of prior model (..) -> the bias of the prior model (..)
- pg 4: To tackle with this difference, we carry out dimensionality reductions in U-Net (..) -> we carry out a dimensionality reduction in the U-Net (..)
- pg 9: The findings above also holds for the (..) -> The findings above also hold for (..)


**Strength And Weaknesses:**

Strengths:
- Clear algorithm derivation, with a very clear exposition of its individual components which make the contribution of the authors very clear.
- Strong relation of the algorithm design to the roots in the physical process itself, hence making this a true physical process-driven design.
- Highly intriguing approach to keep the neural operator's dynamics faithful to the dynamics of the physical process. An approach which should also carry over to similar problems in scientific machine learning
- Strong performance of the algorithm across the presented benchmarks, clearly outperforming previous results.

Weaknesses:
- Missing discussion of the training costs of the proposed algorithm, which make it hard to gauge the extensibility of the algorithm to larger, more complex problems.
- No clear ablations of the influence that each of the 3 operators has on the general performance of the approach. While not immediately clear to the reviewer if the algorithm could be decomposed as such, it would greatly improve the technical depth of the paper if the influences of each operator on the performance could be discerned.
- Evaluations are restricted to the Lorenz-63, and the Lorenz-96 system. Evaluation on more difficult and/or non-Lorenz systems would be of great help to evaluate its applicability to other chaotic systems.

**Summary Of The Paper:**

The work presents an extension to the 4D-Var approach to data assimilation geared towards application on climate data based on neural operator learning. To do so the authors split the neural operator into three distinct operators, the flow-operator, the perturbation-operator, and the observation operator. With a checker board-style training algorithm the individual operators are trained and then tested on two Lorenz-benchmarks.

**Summary Of The Review:**

The authors present a compelling extension to existing 4-D Var data assimilation approaches with a neural operator learning framework, which splits its operators into three distinct operators fulfilling different complementary functions. Overall a strong paper with a very clear exposition, and presenting a novel approach to operator learning for data assimilation, and showing strong performance on Lorenz-systems, the paper would benefit from more ablation analyses, examination across a wider spectrum of benchmarks, as well as at times more references to adjacent literature such as iterative refinement, and similar training algorithms.

---

> ### Author Response · Authors · 2022-11-18
> **Response to Reviewer 1MTc (1/4)**
>
> We sincerely appreciate your careful and thoughtful comments and time. We try to explain your concerns point by point.
>
> **Q1: Training costs**
>
> **A1:** The training cost of our algorithm is affordable on one TITAN RTX GPU, and is also much lower than that in [1].
>
> We report the detailed training cost in the two tables below (the first one is for Lorenz-63 and the second one is for Lorenz-96). If the validation loss does not continue to decrease, we consider the network to converge and end the current phase. The training procedure of [1] is divided into two phases: the first phase is trained with 10 blocks and the second phase is trained with 20 blocks. We should at least allocate 20 epochs for the first phase and 100 epochs for the second phase to make the training loss converge well. As for our work, the numbers of epochs for the phases in the tables correspond to those used in our paper.
>
> In the experiment of the Lorenz-63 system, our network takes approximately 70 minutes to converge, and [1] takes over 200 minutes. As for the Lorenz-96 system, the training time of our network is about 187 minutes, and the training time of [1] is over 520 minutes. We will include the results in our revised version.
>
> Training time for the Lorenz-63 system:
>
> |                          | Phase                          | Training time per epoch | # of epochs | Phase time |
> | ------------------------ | ------------------------------ | ----------------------- | ----------- | ---------- |
> | Fablet et al. (2021) [1] | Phase 1 (10 blocks)            | 1 m 38 s                | 20          | 32 m 40 s  |
> |                          | Phase 2 (20 blocks)            | 1 m 45 s                | 100         | 175 m      |
> |                          | Total time                     |                         |             | 207 m 40 s |
> | Our work                 | Train the observation operator | 7 s                     | 20          | 2 m 20 s   |
> |                          | Train the flow operator        | 10 s                    | 200         | 33 m 20 s  |
> |                          | Finetune 1 (4 blocks)          | 24 s                    | 20          | 8 m        |
> |                          | Finetune 2 (6 blocks)          | 36 s                    | 20          | 12 m       |
> |                          | Finetune 3 (8 blocks)          | 44 s                    | 20          | 14 m 40 s  |
> |                          | Total time                     |                         |             | 70 m 20 s  |
>
> Training time for the Lorenz-96 system:
>
> |                      | Phase                          | Training time per epoch | # of epochs | Time       |
> | -------------------- | ------------------------------ | ----------------------- | ----------- | ---------- |
> | Fablet et al. (2021) | Phase 1 (10 blocks)            | 2 m 34 s                | 20          | 51 m 20 s  |
> |                      | Phase 2 (20 blocks)            | 4 m 45 s                | 100         | 475 m      |
> |                      | Total time                     |                         |             | 526 m 20 s |
> | Our work             | Train the observation operator | 9 s                     | 20          | 3 m        |
> |                      | Train the flow operator        | 33 s                    | 200         | 110 m      |
> |                      | Finetune 1 (4 blocks)          | 33 s                    | 20          | 11 m       |
> |                      | Finetune 2 (6 blocks)          | 48 s                    | 20          | 16 m       |
> |                      | Finetune 3 (8 blocks)          | 63 s                    | 20          | 21 m       |
> |                      | Finetune 4 (10 blocks)         | 78 s                    | 20          | 26 m       |
> |                      | Total time                     |                         |             | 187 m      |
>
> Our framework can adopt more complicated network structures with higher fitting capabilities, such as [7], to scale to larger, more complex problems. More GPU cards can be used in parallel to carry the increased computational cost of larger problems.

---

> ### Author Response · Authors · 2022-11-18
> **Response to Reviewer 1MTc (2/4)**
>
> **Q2: Ablations of the influence of the 3 operators**
>
> **A2:** In our work, the observation operator can be separated from the three operators because we simply use it to map the observations to physical states, thus it is acceptable to use the observation operator alone for reconstruction. The combination of the perturbator and the flow operator aims to refine physical states. They cannot be separated owing to the training issue.
>
> As a result, we add two ablation experiments to show that 1) the refining process with the perturbator and the flow operator greatly contributes to reducing the reconstruction error of the states generated by the observation operator; 2) when the experimental setting for observations is changed, only the observation operator needs retraining, and the network parameters for the perturbator and the flow operator can remain unchanged.
>
> *Experiment 1:* We remove all the perturbators and flow operators, and test the performance of one single observation operator. It has been found that the checkerboard-style refining process can further reduce the reconstruction errors for Lorenz systems by at least 50%, compared with one single observation operator. The detailed results are shown in the table below.
>
> | Dynamic system | Network structure             | R-score | LossDyn  |
> | -------------- | ----------------------------- | ------- | -------- |
> | Lorenz-63      | Observation operator only     | 1.95    | >1.00e-2 |
> |                | Our work with three operators | 0.751   | 1.52e-4  |
> | Lorenz-96      | Observation operator only     | 0.763   | >1.00e-2 |
> |                | Our work with three operators | 0.367   | 2.36e-4  |
>
> *Experiment 2:* Taking the Lorenz-96 system as an example, we design two different settings for observations. For the first setting, the observations are only a random selection of half of the components in the Lorenz system and we sample the observations every four steps; For the second setting, the observations are sampled every four steps for the components whose indexes are even. We conduct the experiments according to the following steps.
>
> Step 1: Train the framework with all three operators under the first experimental setting. We use it as a well-trained reference model.
>
> Step 2: Only train the observation operator under the second experimental setting. We use it to simulate the case where only the observation operator is trained after the observation changes
>
> Step 3: Concatenate the observation operator trained in Step 2 with the perturbator and the flow operator trained in Step 1 without further training.
>
> Step 4: Test the combined network under the second experimental setting.
>
> The results are reported in the following table. Meanwhile, we also report the results of another experiment, in which the observation settings are switched, and the same experiment procedure is conducted again with the above steps. It can be seen that when the perturbator and the flow operator are transferred to another observation setting, they still work well, despite a slight increase in the reconstruction error (smaller than 10%). It also shows that the function of the observation operator and that of the perturbator and the flow operator can be decomposed well.
>
> | Observation setting | In which observation setting the perturbator and the flow operator are trained | R-score | LossDyn |
> | ------------------- | ------------------------------------------------------------ | ------- | ------- |
> | Random              | Random                                                       | 0.367   | 2.36e-4 |
> |                     | Even                                                         | 0.402   | 3.61e-4 |
> | Even                | Random                                                       | 0.304   | 3.61e-4 |
> |                     | Even                                                         | 0.291   | 2.56e-4 |

---

> ### Author Response · Authors · 2022-11-18
> **Response to Reviewer 1MTc (3/4)**
>
> **Q3: Evaluation on non-Lorenz systems**
>
> **A3:** We add one more experiment for the Vissio-Lucarini20 [2] system. It is a coupled system intended to represent a minimal model of the earth's atmosphere, which is composed of two types of variables, namely, kinetic variables and thermodynamical variables. Since both of the variables are coupled, this system is more complicated than the Lorenz systems.
>
> *Experimental settings:* The numbers of both kinetic variables and thermodynamical variables are set to 40. The observations are sampled every four steps for a random half of both kinds of variables. The variance of Gaussian noise for observations is 2.0.
>
> *Network design:* In our proposed method, we adopt a three-layer U-Net similar to that in the Lorenz-96 system. The difference is that we change the channel number from one to two, in order to fit the characteristics of the system with two types of variables. As for the baseline, [1] does not implement their neural network for the Vissio-Lucarini20 system, and we do our best to reproduce their algorithm. The network we use is similar to the one that can reproduce the results in [1] for the Lorenz-96 system, which adds confidence to our implementation of [1] for the Vissio-Lucarini20 system.
>
> *Experimental results:* Similar to the results in the Lorenz-96 system, our network outperforms [1] in terms of both R-score, dynamic loss, and test time, as shown in the table below. Test time may vary across experiments, and we provide the average and standard deviation of test-time results from 10 independent tests.
>
> | Dynamic system        | Model                | R-score | LossDyn | AVG (s) | SD (s) |
> | --------------------- | -------------------- | ------- | ------- | ------- | ------ |
> | Vissio-Lucarini20 [2] | Our work (10 blocks) | 0.352   | 3.57e-4 | 23.6    | 1.5    |
> |                       | Fablet et al. (2021) | 0.358   | 3.25e-2 | 64.2    | 3.4    |
>
> Training time for the Vissio-Lucarini20 system:
>
> |                      | Phase                          | Training time per epoch | # of epochs | Phase time |
> | -------------------- | ------------------------------ | ----------------------- | ----------- | ---------- |
> | Fablet et al. (2021) | Phase 1 (10 blocks)            | 2 m 40 s                | 20          | 53 m 20 s  |
> |                      | Phase 2 (20 blocks)            | 4 m 57 s                | 100         | 495 m      |
> |                      | Total time                     |                         |             | 548 m 20 s |
> | Our work             | Train the observation operator | 22 s                    | 20          | 7 m 20 s   |
> |                      | Train the flow operator        | 34 s                    | 200         | 113 m 20s  |
> |                      | Finetune 1 (4 blocks)          | 57 s                    | 20          | 19 m       |
> |                      | Finetune 2 (6 blocks)          | 81 s                    | 20          | 27 m       |
> |                      | Finetune 3 (8 blocks)          | 105 s                   | 20          | 35 m       |
> |                      | Finetune 4 (10 blocks)         | 129 s                   | 20          | 43 m       |
> |                      | Total time                     |                         |             | 224 m 40 s |

---

> ### Author Response · Authors · 2022-11-18
> **Response to Reviewer 1MTc (4/4)**
>
> **Q4: How the learning procedure relates to the previous works**
>
> **A4:** We have noticed the major works on diffusion models [3,4,5] and optimizer learning [6], both of which employ the iterative learning procedure to refine the generated outputs.
>
> *Relationship with diffusion models:*  The diffusion models define a Markov chain of diffusion steps to slowly add random noise to the data and then learn to reverse the diffusion process to construct desired data samples from the noise. The advantage of the diffusion models is that they generate images with high quality and exhibit good interpretability under the Bayesian inference framework. We will consider combining our framework with diffusion models to improve interpretability and accuracy in our future work.
>
> *Relationship with optimizer learning:* Optimizer learning refers to a group of algorithms that builds neural networks, especially recurrent neural networks, to learn an optimizer for a certain set of objective functions. Both the optimizer learning and our proposed method progressively refine the outputs step by step. The major differences between our models and optimizer learning are summarized as follows: In optimizer learning, the blocks in neural networks are generally constructed based on the theory of gradient descent, whereas the three operators in our model are inspired by the characteristics of the data assimilation problem. As a result, optimizer learning algorithms are applicable to all optimization problems theoretically, while our method is mainly applicable to the data assimilation problem. Meanwhile, for data assimilation, our method outperforms the optimizer learning-based baseline [1], as shown in our paper.
>
> **Q5: Discussion of the choice of using the U-Net architecture**
>
> **A5:** We really appreciate your recommendation. We have read the work of Wang et al. [7], and we think that the idea of three-level decomposition in this paper is very novel and attractive. In future work, we are willing to try this network architecture in our framework when we build our model on more complex systems, especially those related to turbulent flow. Also, we are willing to include the discussion of the work of Wang et al. in the network design part of our revised version.
>
> **Q6: Typos**
>
> **A6:** Thank you for kindly pointing out the typos. We will correct them in the revised version of our paper.
>
> [1] Fablet, Ronan, et al. "Learning variational data assimilation models and solvers." *Journal of Advances in Modeling Earth Systems* 13.10 (2021): e2021MS002572.
>
> [2] Vissio, Gabriele, and Valerio Lucarini. "Mechanics and thermodynamics of a new minimal model of the atmosphere." *The European Physical Journal Plus* 135.10 (2020): 1-21.
>
> [3] Sohl-Dickstein, Jascha, et al. "Deep unsupervised learning using nonequilibrium thermodynamics." *International Conference on Machine Learning*. PMLR, 2015.
>
> [4] Ho, Jonathan, Ajay Jain, and Pieter Abbeel. "Denoising diffusion probabilistic models." *Advances in Neural Information Processing Systems* 33 (2020): 6840-6851.
>
> [5] Tashiro, Yusuke, et al. "CSDI: Conditional score-based diffusion models for probabilistic time series imputation." *Advances in Neural Information Processing Systems* 34 (2021): 24804-24816.
>
> [6] Andrychowicz, Marcin, et al. "Learning to learn by gradient descent by gradient descent." *Advances in neural information processing systems* 29 (2016).
>
> [7] Wang, Rui, et al. "Towards physics-informed deep learning for turbulent flow prediction." *Proceedings of the 26th ACM SIGKDD International Conference on Knowledge Discovery & Data Mining*. 2020.

---

> > ### Comment · Reviewer_1MTc · 2022-11-19
> > **Addressing of Concerns**
> >
> > I sincerely thank the authors for the thoughtful and in-depth treatment of may questions.Seeing some of my concerns addressed, e.g. performance on a non-Lorenz system, I will raise my score.

---

### Decision · Program_Chairs · 2023-01-20

**Decision:**

Reject

**Justification For Why Not Higher Score:**

Many inaccuracies and missing information in the document.
Lack of positioning in relation to conventional methods.

**Justification For Why Not Lower Score:**

N/A

**Metareview: Summary, Strengths And Weaknesses:**

The paper introduces a machine learning method for the problem of data assimilation with the objective of reducing the computational cost compared to classical approaches like 4D-Var. Data assimilation is the reconstruction of the sequence of the hidden states of a dynamical system given a series of observations that provide incomplete information of the states via an observation operator.   It is widely used e.g. in numerical weather prediction and for many other problems. The authors start from a variational formulation of the problem involving two loss terms: a reconstruction term measures how well the states are estimated and a dynamical term measures how close are the empirical and the theoretical prior dynamics. They consider a supervised setting were the system state at some space-time points is available. This is a different assumption than the classical 4D-Var algorithm that opens the way to a different and simpler optimization strategy. The proposed method relies on three components, each implemented through a neural network that respectively inverse the observation operator, learn the dynamics and the state reconstruction.  Training is performed by first learning the inverse observation operator for computing a first guess of the state vector and then progressively training the elements responsible for the state reconstruction and dynamics that will refine this first guess. Experiments and comparisons with a baseline are performed on two Lorenz systems.

The authors tackle an important problem since assimilation is a key component for the incorporation of observations in the modeling of earth systems. This work follows a recent trend exploring the use of data-based methods for improving or complementing classical methods. The proposed method performs well compared to the baseline used in the paper. Thanks to the supervised setting, it allows the model to directly optimize the reconstruction term and to perform assimilation on a whole assimilation window at once, thus providing a speedup w.r.t traditional techniques. The authors answered with extensive responses to the reviewers’ questions and added new experiments. However, I have two reservations about accepting this article. One concerns the positioning w.r.t. the classical setting of assimilation. The supervised assumption used here greatly simplifies the assimilation problem and makes the proposed solution relatively straightforward. This also implies that the solution cannot be compared with traditional approaches which do not assume the availability of such a supervision. The second one concerns the technical description that despite clear improvements compared with the first version of the paper remains too vague with several assumptions, concepts and technical details not clearly specified. There are even some notation problems that remain. Note also that the presentation to an ML audience would benefit from a short but precise introduction to variational data assimilation. In its present form the paper does not allow to fully appreciate the significance of the contribution and will not benefit as it probably could to the audience.